# Rapid, adaptable and sensitive Cas13-based COVID-19 diagnostics using ADESSO

Beatrice Casati [1,2], Joseph Peter Verdi[1,3], Alexander Hempelmann[3], Maximilian Kittel[4,5],
Andrea Gutierrez Klaebisch[5,6], Bianca Meister[5,6], Sybille Welker[5,6], Sonal Asthana[1], Salvatore Di Giorgio [1],
Pavle Boskovic[7], Ka Hou Man[7], Meike Schopp[8], Paul Adrian Ginno[8], Bernhard Radlwimmer[7],
Charles Erec Stebbins[3], Thomas Miethke [5,6,9✉], Fotini Nina Papavasiliou [1,9✉] & Riccardo Pecori [1,9✉]

During the ongoing COVID-19 pandemic, PCR testing and antigen tests have proven critical for helping to stem the spread of its causative agent, SARS-CoV-2. However, these methods suffer from either general applicability and/or sensitivity. Moreover, the emergence of variant strains creates the need for flexibility to correctly and efficiently diagnose the presence of substrains. To address these needs we developed the diagnostic test ADESSO (Accurate Detection of Evolving SARS-CoV-2 through SHERLOCK (Specific High Sensitivity Enzymatic Reporter UnLOCKing) Optimization) which employs Cas13 to diagnose patients in 1 h without sophisticated equipment. Using an extensive panel of clinical samples, we demonstrate that ADESSO correctly identifies infected individuals at a sensitivity and specificity comparable to RT-qPCR on extracted RNA and higher than antigen tests for unextracted samples. Altogether, ADESSO is a fast, sensitive and cheap method that can be applied in a point of care setting to diagnose COVID-19 and can be quickly adjusted to detect new variants.

---

[1] Division of Immune Diversity, Department of Immunology and Cancer, German Cancer Research Centre (DKFZ), 69120 Heidelberg, Germany. [2] Faculty of Biosciences, Heidelberg University, 69120 Heidelberg, Germany. [3] Division of Structural Biology of Infection and Immunity, Department of Immunology and Cancer, German Cancer Research Centre (DKFZ), 69120 Heidelberg, Germany. [4] Institute of Clinical Chemistry, Medical Faculty of Mannheim, University of Heidelberg, Theodor-Kutzer-Ufer 1-3, 68167 Mannheim, Germany. [5] Mannheim Institute for Innate Immunoscience (MI3), Medical Faculty of Mannheim, University of Heidelberg, Ludolf-Krehl-Str. 13–17, 68167 Mannheim, Germany. [6] Institute of Medical Microbiology and Hygiene, Medical Faculty of Mannheim, University of Heidelberg, Theodor-Kutzer-Ufer 1-3, 68167 Mannheim, Germany. [7] Division of Molecular Genetics, German Cancer Research Centre (DKFZ), 69120 Heidelberg, Germany. [8] Division of Regulatory Genomics and Cancer Evolution, German Cancer Research Centre (DKFZ), Heidelberg, Germany. [9] These authors jointly supervised this work: Thomas Miethke, Fotini Nina Papavasiliou, Riccardo Pecori. ✉email: Thomas.Miethke@medma.uni-heidelberg.de; n.papavasiliou@dkfz.de; r.pecori@dkfz.de

Since the beginning of the coronavirus disease 2019 (COVID-19) pandemic, 486 million confirmed cases, including 6.1 million deaths, have been reported globally as of April 3, 2022[1]. COVID-19 is a respiratory disease caused by severe acute respiratory syndrome coronavirus 2 (SARS-CoV-2)[2,3]. The quick diffusion of SARS-CoV-2 is primarily attributed to the relatively long duration of viral shedding by infected individuals, the viral load dynamics and the lengthy incubation period of 5-6 days[4–6]. The viral load peaks soon after the onset of symptoms[7–9], suggesting that individuals with COVID-19 begin viral shedding a few days before symptoms appear. Further, a significant proportion of infected individuals either remain entirely asymptomatic or only manifest mild symptoms[6,10], thus facilitating the spread of the virus and leading to the current pandemic.

The situation has been exacerbated by the fact that SARS-CoV-2 has evolved considerably. The first variant to appear carried a D614G mutation in the spike protein[11] which is now dominant and shared between all existing variants. The virus has since accumulated multiple additional mutations in varying combinations, resulting in more transmissible and potentially more virulent variants threatening several countries worldwide.

The urgent need for a prophylactic response has accelerated the development of multiple effective vaccines and more than 11 billion doses have been administered globally[1]. However, even in the most optimistic scenario, it will take time to reap the benefits of a global vaccination campaign. This is especially relevant in low-income countries where the vaccination rate is dramatically lower[12]. Therefore, complementary efforts to limit the spread of the virus are still essential. A recent model of viral dynamics indicates that frequent testing is essential for efficient identification and isolation of carriers and containment of the pandemic[13]. A sensitive, accurate, accessible and reliable test with short turnaround time is thus highly desirable.

The worldwide gold standard diagnostic test for SARS-CoV-2 infection is the reverse transcription-quantitative polymerase chain reaction (RT-qPCR). While sensitive and effective, it comes with the important limitation of requiring specific equipment, laboratory infrastructures and qualified personnel. Inadequate access to such resources significantly reduces the frequency of testing. Additionally, PCR testing facilities often require days to report the test outcome, resulting in a long sample-to-result turnaround time. To face these challenges, different rapid tests have been implemented, such as rapid PCR and antigen-based tests. While these advancements represent significant progress in diagnosing COVID-19, rapid PCR tests still require specific equipment[14] and antigen-based tests have lower sensitivity and specificity[15,16]. In fact, multiple investigations on the accuracy and reliability of antigen-based tests have concluded that their use is beneficial only for the detection of infected individuals with high viral titers[17,18]. Taken as a whole, there is still a need for an alternative test that is comparable to RT-qPCR in terms of sensitivity and specificity, yet faster and independent of complex instruments.

CRISPR diagnostic (CRISPR-Dx) technologies offer promising solutions to meet all these requirements[19]. The CRISPR bacterial system is capable of recognizing and cleaving foreign genetic material. Among the CRISPR associated (Cas) proteins, Cas13 and Cas12 specifically bind RNA and DNA molecules, respectively, complementary to the target-binding CRISPR RNA (crRNA). Upon target recognition, the Cas proteins cleave a reporter in *trans*, which can then be detected via different readouts[20–22]. To achieve high sensitivity, isothermal amplification methods that do not rely on sophisticated equipment, such as loop-mediated isothermal amplification (LAMP)[23] or recombinase polymerase amplification (RPA)[24], have been combined with Cas-mediated nucleic acid detection[21,25]. CRISPR-Dx

technologies were quickly adapted for the detection of SARS-CoV-2[26–36] and two of them have received emergency use authorization from the Food and Drug Administration (FDA), with use restricted to the approved laboratories[37,38]. Despite their high potential, most of these technologies require either extracted RNA[29,32–35] or show a higher limit of detection (LoD) when performed on unextracted samples[26,28]. Finally, while CRISPR-Dx technologies were benchmarked against RT-qPCR, the analysis of their performance on clinical samples in direct comparison with antigen tests is lacking.

Here we have optimized the Cas13a-based "SHERLOCK"[25] platform to develop ADESSO (Accurate Detection of Evolving SARS-CoV-2 through SHERLOCK Optimization). ADESSO demonstrates highly sensitive detection of SARS-CoV-2 and its variants directly from patient-derived material. The entire protocol is completed in approximately one hour, does not require RNA extraction or any specific equipment, has a detection limit of 2.5 cp/μl of SARS-CoV-2 synthetic genome (approaching the limit of RT-qPCR) and is low-cost (less than 5€ per test). Throughout our work, we extensively evaluated the real diagnostic potential of ADESSO in direct comparison to RT-qPCR and antigen testing on samples collected with two different methods (nasopharyngeal swab (NP) and gargle of saline). To ensure that our sample cohort represented a relevant portion of the population that can remain undetected, we included ambulatory patients with minimal or mild symptoms as well as asymptomatic individuals who had recently been in contact with COVID-19 positive patients. Our study showed that ADESSO has a sensitivity comparable to RT-qPCR when applied to purified RNA. When employed directly on unextracted samples, ADESSO outperformed the rapid antigen test, demonstrating its potential as a more sensitive and reliable point of care (POC) diagnostic test. Furthermore, we adapted ADESSO to detect specific mutations characteristic of four different SARS-CoV-2 variants of concern (VOCs) and successfully demonstrated the specificity of each variant-tailored ADESSO test in clinical samples.

## Results

**ADESSO: an optimized and highly-sensitive SHERLOCK assay.** We began developing ADESSO by determining the sensitivity of the Cas13-based SHERLOCK (Specific High Sensitivity Enzymatic Reporter UnLOCKing)[25] platform on clinical samples (Supplementary Fig. 1). To increase sensitivity and reduce duration of the assay, we evaluated alternative reagents and different reaction conditions for RNA extraction, the isothermal amplification of viral RNA via RT-RPA and detection of a specific RNA sequence by Cas13 (Supplementary Fig. 1a). We assessed Cas13 activation with a fluorometer to monitor the speed of the reaction in real-time and via a lateral flow-based visual readout as an instrument-free detection method that would be used in a POC test. The fluorescence and lateral flow readouts are based on the use of two different RNA reporters, where a fluorophore (e.g., FAM) is flanked by either a quencher (for fluorescence readouts) or biotin (for lateral flow readouts). Upon Cas13-mediated cleavage of the reporter, the fluorophore is cut from either the quencher or biotin. For fluorescence-based readouts, cleavage of the RNA reporter releases the fluorophore from the quencher and the fluorescent signal can then be detected by a fluorometer (Supplementary Fig. 1a). In the lateral flow scenario, the resulting signal can be read on a lateral flow strip where gold-labeled antibodies against FAM are used to visualize the reporter. In a negative sample, the RNA reporter flanked by FAM and biotin is intact and is captured by a first line of streptavidin resulting in a band called "control band". In a positive sample, the reporter is cut, releasing the FAM-containing fragment to be captured by a

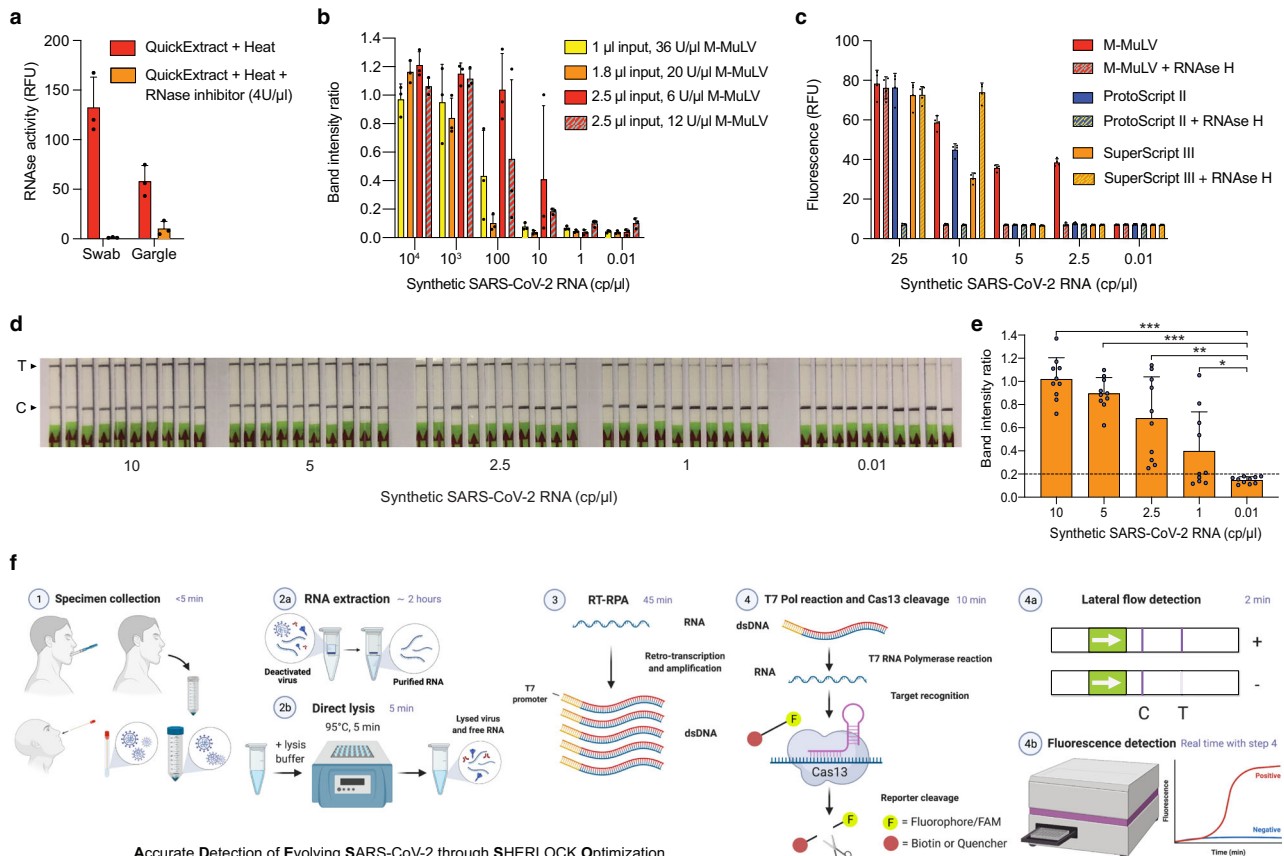

**Fig. 1 ADESSO: an optimized and highly sensitive SHERLOCK-based assay for SARS-CoV-2 detection. a** Measurement of RNase activity in a swab and gargle sample lysed at 95 °C for 5 minutes with QuickExtract DNA Extraction Solution enriched or not with Murine RNase inhibitor, at a final concentration of 4 U/μl. **b** Determination of sensitivity on serial dilutions of SARS-CoV-2 synthetic genome upon optimization of RT units and input volume in the RT-RPA reaction with lateral flow readout. **c** Sensitivity on serial dilutions of SARS-CoV-2 genome using different reverse transcriptases in presence or absence of RNase H. **d** Sensitivity of the improved protocol with lateral flow readout on serial dilutions of SARS-CoV-2 synthetic genome upon integration of all the above-described optimizations. **e** Intensity ratios of the lateral flow strips in **d** are shown in a bar plot. An unpaired, two-tailed t-test was performed (*$P = 0.0302$; **$P = 0.0002$; ***$P < 0.0001$). **f** Graphic of the experimental workflow of ADESSO to detect SARS-CoV-2, with or without RNA extraction, in clinical samples with lateral flow or fluorescence readout. For panels **d**, **f**: T = test band; C = control band. For panels **a**, **b**, **c**, **e**: bars represent the mean for n replicates and error bars represent the positive standard deviation. For **a**, **b**: $n = 3$ biological replicates; for **c**: $n = 4$ technical replicates; for **e**: $n = 10$ biological replicates.

second line of antibodies resulting in a "test band". The band intensity ratio between the test band and the control band thus reflects the level of Cas13 activation (Supplementary Fig. 1a). Positive samples were empirically determined to be represented by a band intensity ratio higher than 0.2. This threshold was defined based on the band intensity ratio obtained in all the negative controls and samples used in this study ($n = 282 + 400$; Supplementary Fig. 2) and is in agreement with previously published findings[30].

The RNA extraction step is a time-consuming, labor-intensive and costly step for COVID-19 diagnosis (Supplementary Fig. 1a). It has furthermore been complicated by the global shortage of RNA extraction kits throughout the pandemic[39]. Therefore, we optimized the SHERLOCK method parameters to allow for circumvention of this step while maintaining high sensitivity. We first minimized RNase activity during sample lysis by adding RNase Inhibitors in the lysis buffer. After a 5 min incubation at 95 °C in QuickExtract DNA Extraction solution (as previously shown[26]) in the presence or absence of RNase Inhibitors, RNaseAlert was added to the sample to evaluate nuclease activity and fluorescence was measured. Notably, the addition of RNase inhibitors in the lysis buffer prior to heating was sufficient to inhibit RNase activity almost completely for both swab and

gargle samples (Fig. 1a). Next, we optimized the amount of RT units and the volume of sample input in the RT-RPA step using dilutions of the synthetic SARS-CoV-2 genome spiked into a mixture of a negative swab sample, RNase inhibitor and QuickExtract. We observed the best results with 6 U/μl of RT and 2.5 μl of sample input per reaction, demonstrating the importance of sample input over RT units to achieve higher sensitivity (Fig. 1b). Additionally, commercially available RT enzymes have weak or no RNase H activity. Thus, to further optimize the RT-RPA step we compared different RT enzymes in the presence or absence of RNase H. M-MuLV showed the best sensitivity (5–2.5 cp/μl) in comparison to ProtoScript II or SuperScript III, while the addition of RNase H led to an improvement for SuperScript III only (Fig. 1c). Secondly, we tested varying final concentrations of RPA, where 1xRPA corresponds to the standard amount of RPA described in the original SHERLOCK protocol[25,40] and 5xRPA corresponds to the optimal amount according to the manufacturer's instructions. To assess this, we performed our assay with different concentrations of RPA on a positive clinical sample with Ct = ~29 (Supplementary Table 1), which is approximately the LoD of other Cas13-based tests[28,30]. Remarkably, while we obtained a false negative with 1xRPA, the sample resulted positive for final concentrations

of RPA from 2× to 5× (Supplementary Fig. 3a, b). Bearing in mind the cost per single test, we decided to proceed using a 2xRPA concentration. Finally, in order to optimize the Cas13 detection step, we made a ten-fold dilution of a positive RT-RPA reaction (50 cp/μl) and we performed Cas13 detection using the original concentration of Cas13/crRNA (45/22.5 nM)[25,40], in comparison to higher amounts (Supplementary Fig. 3c). A concentration of Cas13/crRNA of 90 nM each led to a faster reaction, reaching the plateau after 15 min only, compared to 30 min for the other two concentrations (Supplementary Fig. 3d). We also confirmed that a 10 min incubation for Cas13 detection is sufficient to yield a clearly positive outcome in the lateral flow detection assay (Supplementary Fig. 3e, f), which is an essential feature for a POC test. Moreover, a shorter Cas13 reaction allows us to extend the incubation time of the RT-RPA step for highly sensitive reactions[40] without affecting the total time of the assay. Finally, we assessed the sensitivity of this optimized protocol on serial dilutions of SARS-CoV-2 synthetic genome and we observed a significant reproducible sensitivity of 2.5 cp/μl (Fig. 1d, e). We named this new optimized diagnostic assay ADESSO (Accurate Detection of Evolving SARS-CoV-2 through SHER-LOCK Optimization) (Fig. 1f).

**Evaluation of ADESSO performance on clinical samples**. We used ADESSO to test a total of 195 clinical samples in direct comparison to RT-qPCR and an antigen test commonly used for the diagnosis of COVID-19[41]. To allow a fair comparison between the methods, we first selected 95 positive and 100 negative individuals (via COBAS RT-qPCR on NP swab). For each of these specimens, RNA was re-extracted and analyzed by both RT-qPCR (Tib Molbiol) and ADESSO using a lateral flow readout to simulate a POC test. Additionally, an antigen-based diagnostic test (RIDA QUICK SARS-CoV-2-Antigen) and ADESSO were performed directly on unextracted samples. Finally, we also obtained saline gargle specimens from the same individuals as an alternative sampling method (Fig. 2a).

This randomly selected cohort of positive individuals covers the full distribution of viral titers between Ct 17 and Ct 38 (Fig. 2b, c and Supplementary Table 2)[42], thus allowing us to avoid bias during LoD evaluation due to sample size or viral load distribution[43]. The results of this experiment are summarized in Table 1. ADESSO on RNA extracted from swabs was able to correctly identify most positive samples (91 out of 95), resulting in a sensitivity of 96% and a LoD corresponding to Ct value ~32 (Fig. 2b, Supplementary Fig. 4a). RT-qPCR (Tib Molbiol) performed on the same samples was largely in agreement with the COBAS RT-qPCR, with highly correlated Ct values (Supplementary Fig. 4e). Using this method, we were able to identify 89 out of 95 positive samples, resulting in a slightly lower sensitivity of 94% compared to ADESSO (Table 1 and Fig. 2b, Supplementary Fig. 4e). This side-by-side testing demonstrates the accuracy of ADESSO when performed on extracted RNA. Although ADESSO on unextracted swab samples resulted in a reduced sensitivity (77%) and LoD (Ct value 30), it still strikingly outperformed the antigen test, which only detected 44 out of 95 positive samples, resulting in a sensitivity of 46% and a LoD corresponding to Ct 23 (Table 1 and Fig. 2b, Supplementary Fig. 4b). Additionally, while ADESSO and the RT-qPCR test showed 100% specificity, we observed one false positive sample by antigen test (Table 1). Similar results were observed on saline gargle samples with a general drop in sensitivity and LoD for all the detection methods (Table 1 and Fig. 2c, Supplementary Fig. 4c,d). Interestingly, this drop seems to be related to the sampling procedure. Indeed, higher RT-qPCR Ct values were observed in gargle specimens compared to their matched swab

samples (Supplementary Fig. 4e–g). This reduction was even more pronounced for the antigen test and it might explain why the manufacturers only recommend using nose and/or throat swabs for the execution of the antigen test[44]. A similar drop in sensitivity was also reported in other studies where paired nasopharyngeal swab-saliva samples were tested[45,46]. Altogether, ADESSO demonstrated similar sensitivity to RT-qPCR (Tib Molbiol) on extracted RNA and outperformed the antigen test when performed on unextracted samples. These results validate the high potential of ADESSO as a POC test for the detection of SARS-CoV-2 infected individuals.

**Adaptation of ADESSO for detection of SARS-CoV-2 variants**. In order to demonstrate that ADESSO can be easily adapted for the detection of SARS-CoV-2 variants, we focused our attention on four of the identified SARS-CoV-2 variants of concern (VOCs): Alpha (B.1.1.7), Beta (B.1.351), Delta (B.1.617.2) and Omicron (B.1.1.529). SARS-CoV-2 VOCs are characterized by genetic changes affecting at least one of the following viral features: transmissibility, disease severity, interaction with the host immunity, response to social measures or available diagnostics, vaccines or treatments[47–51]. To demonstrate the adaptability of ADESSO we selected specific mutations for each VOC and we designed crRNAs to recognize these altered sequences. We selected the deletion ΔHV69-70 for the Alpha strain and the mutations D80A and T478K for Beta and Delta, respectively. For Omicron we chose A67V in combination with the ΔHV69-70 deletion. Since the selected mutations for Alpha, Beta and Omicron are located in close proximity, they can be detected by amplifying one single region of the SARS-CoV-2 S gene with the same primer pair during the RT-RPA reaction, while the mutation specific for Delta is located in a different region of the S gene and required alternate primers (Fig. 3a). A crRNA specific to each variant is subsequently used in a 20-min Cas13 reaction to differentiate between VOCs (Fig. 3a and Supplementary Fig. 5a, b). We named each specific test by the variant it identifies: ADESSO-Alpha, -Beta, -Delta and -Omicron (Fig. 3a). The need for a 20-min Cas13 reaction for the variant-specific ADESSO compared to the 10-min reaction used for the standard ADESSO is probably due to the few mismatches between crRNAs and target (Fig. 3a). These mismatches are essential to obtain specific detection of single nucleotide variants[25] but at the same time they result in a slower reaction[52]. A previous study showed that single nucleotide variations can be detected designing a crRNA in which the target mutation is in position 3 and two additional synthetic mismatches are placed in positions 5 and 24, highlighting the importance of these three sites for Cas13 activity[25]. While this design was ideal for the crRNAs in ADESSO-Beta and Delta (Fig. 3), ADESSO-Omicron required further optimization to achieve single nucleotide discrimination between Alpha and Omicron samples. Considering that position 7 was also shown to be critical for Cas13 activity[25], we designed a new crRNA in which the target mutation is located in this position, and then we added three synthetic mismatches in positions 3, 5 and 24 (Fig. 3a). This design led to successful single nucleotide discrimination between Alpha and Omicron samples (Fig. 3). This aspect points to the need for a tailored crRNA optimization for some targets. We also observed a slight increase in signal from cross-variant reactions, mostly for ADESSO-Alpha on Omicron samples. Therefore, we increased the band intensity ratio threshold from 0.2 to 0.4 to avoid any false positives during variant identification (Fig. 3b). Finally, we performed a blind test on clinical samples carrying one of the aforementioned variants. The presence of variant strains was previously determined by sequencing the genome of the virus present in those samples

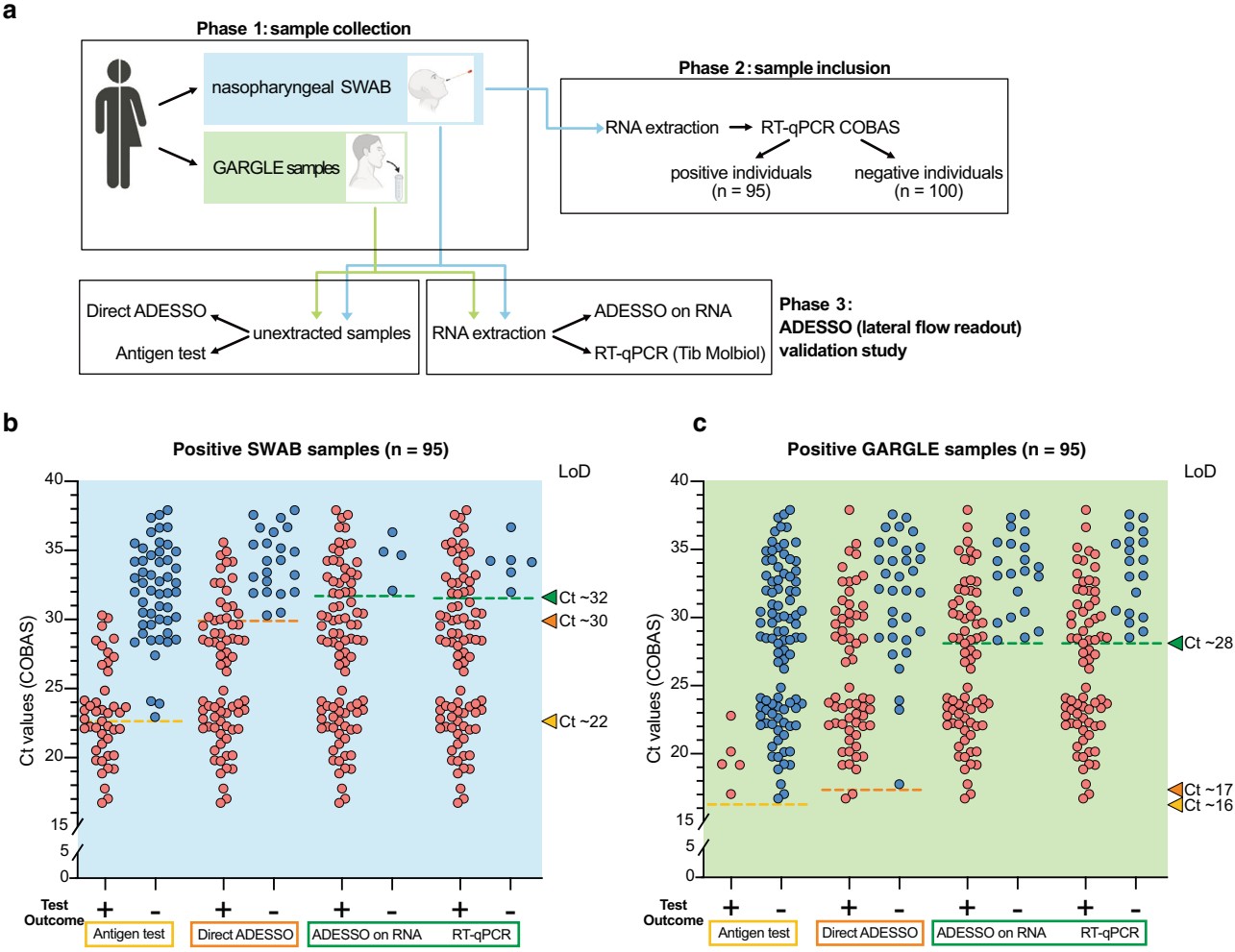

**Fig. 2 Evaluation of ADESSO performance on clinical samples in direct comparison to RT-qPCR and antigen test. a** Schematic of the validation study to assess ADESSO performance for SARS-CoV-2 detection in clinical samples in comparison with RT-qPCR (Tib Molbiol) and antigen test (RIDA QUICK SARS-CoV-2 Antigen). The COVID-19 status of the samples included in the study was initially determined by RT-qPCR (COBAS). ADESSO was performed on both extracted RNA and unextracted samples with lateral flow readout. **b, c** SARS-CoV-2 detection and LoD evaluation in swab (**b**) and gargle (**c**) samples collected from 95 COVID-19 positive individuals using RIDA QUICK SARS-CoV-2 Antigen test, ADESSO and Tib Molbiol RT-qPCR. Each sample is represented by a red or blue dot for positive or negative test outcome, respectively. "Direct" ADESSO and RIDA QUICK SARS-CoV-2 Antigen tests were performed on unextracted samples, while ADESSO and Tib Molbiol RT-qPCR were performed on extracted RNA. The LoD for each test is represented as a dotted line and the corresponding Ct value is indicated by an arrowhead on the right Y axis.

(Supplementary Data 1). Using the variant-specific ADESSO tests we were able to correctly identify all the variants among the clinical samples analyzed (Fig. 3b and Supplementary Fig. 5c). Additionally, using the standard ADESSO we could detect all samples with the exception of V#6–8 (Fig. 3b and Supplementary Fig. 5c). These three Beta samples shared a point mutation (R246I) close to a deletion (Δ242–244) within the sequence bound by the forward primer used in the RT-RPA step of ADESSO (Supplementary Fig. 5d). On one hand, this observation demonstrates that ADESSO is resistant to deletions of several nucleotides within this region because Beta samples carrying only the deletion Δ242–244 were successfully detected (V#9,10; Fig. 3b, Supplementary Fig. 5c, d). On the other hand, samples V#6–8 carrying the additional R246I mutation at the 3' end of the region recognized by the same primer led to a negative test (Supplementary Fig. 5c, d). This last observation is particularly important in the context of the results shown in Fig. 2. It is possible that the aforementioned mutation could have led to some false negatives, affecting the sensitivity of the assay. However, after performing bioinformatic analysis on ~3.7 million SARS-CoV-2 genomes, we

observed that both these mutations (Δ242–244 and R246I) occur at a very low frequency (<0.001) (Supplementary Fig. 5e; Supplementary Data 2–4). Additionally, using the same dataset we have confirmed that ADESSO RT-RPA primers and crRNA show exact target matches in almost 100% of all SARS-CoV-2 genomes analyzed, demonstrating that the performance of ADESSO will very unlikely be affected by the so far identified mutations (Supplementary Fig. 5e; Supplementary Data 2–4). Nonetheless, alternative regions could be explored where an even smaller or null variation is observed to develop an ADESSO assay which is even more robust.

Altogether, these results show how ADESSO can be readily adapted for the detection of SARS-CoV-2 variants of concern and even specific mutations. This feature of our assay is a crucial aspect for the COVID-19 pandemic, where quick identification of known circulating variants is essential to contain their spread.

**Estimation of the infected population detected by ADESSO.** In order to understand what portion of the SARS-CoV-2 infected population ADESSO would be able to detect, we analyzed the Ct

**Table 1 Predictive values, sensitivity and specificity of ADESSO, Tib Molbiol RT-qPCR and RIDA QUICK SARS-CoV-2 antigen test on swab and gargle samples.**

| Sampling method | Sample | Test | Test result | Pos. samples (N = 95) | Neg. samples (N = 100) | Tot. samples (N = 195) | Positive predictive value | Negative predictive value | Sensitivity | Specificity |
|---|---|---|---|---|---|---|---|---|---|---|
| SWAB | RNA | RT-qPCR | Positive | 89 | 0 | 89 | 89/89 (100%) | | 89/95 (94%) | |
| | | | Negative | 6 | 100 | 106 | | 100/106 (94%) | | 100/100 (100%) |
| | | ADESSO | Positive | 91 | 0 | 91 | 91/91 (100%) | | 91/95 (96%) | |
| | | | Negative | 4 | 100 | 104 | | 100/104 (96%) | | 100/100 (100%) |
| | Lysate | ADESSO | Positive | 73 | 0 | 73 | 73/73 (100%) | | 73/95 (77%) | |
| | | | Negative | 22 | 100 | 122 | | 100/122 (82%) | | 100/100 (100%) |
| | | antigen test | Positive | 44 | 1 | 45 | 44/45 (98%) | | 44/95 (46%) | |
| | | | Negative | 51 | 99 | 150 | | 100/150 (67%) | | 99/100 (99%) |
| GARGLE WATER | RNA | RT-qPCR | Positive | 75 | 0 | 75 | 75/75 (100%) | | 75/95 (79%) | |
| | | | Negative | 20 | 100 | 120 | | 100/120 (83%) | | 100/100 (100%) |
| | | ADESSO | Positive | 74 | 0 | 74 | 74/74 (100%) | | 74/95 (78%) | |
| | | | Negative | 21 | 100 | 121 | | 100/121 (83%) | | 100/100 (100%) |
| | Lysate | ADESSO | Positive | 62 | 0 | 62 | 62/62 (100%) | | 62/95 (65%) | |
| | | | Negative | 33 | 100 | 133 | | 100/133 (75%) | | 100/100 (100%) |
| | | antigen test | Positive | 5 | 0 | 5 | 5/5 (100%) | | 5/95 (5%) | |
| | | | Negative | 90 | 100 | 190 | | 100/190 (53%) | | 100/100 (100%) |

value distribution across a population of 6,439 infected individuals among ambulatory patients presenting minimal to mild symptoms as well as asymptomatic people who had contact with COVID positive individuals between October 1, 2020, and July 31, 2021. We observed a distribution ranging from Ct 17 to Ct 38. This observation confirms the fact that also these individuals can manifest high viral loads and therefore be infectious[53,54], thus facilitating the oblivious spread of the virus. Based on the LoD evaluation shown in this study, by applying ADESSO on unextracted swab samples an estimated ~70% of the infected population would be successfully detected (Fig. 4). This portion is remarkably higher (31% more) than the one detected by antigen tests that are currently commonly used in our society. Notably, mathematical models show that successful identification and isolation of 50% of infected individuals is already sufficient to flatten the infection curve[55]. Our test exceeds this fraction in all conditions (Table 1 and Fig. 4), strongly suggesting that immediate and widespread application of ADESSO would be of great help to contain the pandemic.

## Discussion

Here, we describe ADESSO, a Cas13-based optimized method for highly sensitive COVID-19 testing. Overall, we tested 793 samples (393 positive and 400 negative, Supplementary Table 1–4) and compared these results with both a RT-qPCR and antigen test. ADESSO has a sensitivity and specificity comparable to RT-qPCR when performed on RNA extracted from either swabs or gargle samples (Fig. 2 and Table 1). Additionally, in order to evaluate the potential of ADESSO as a POC test, we benchmarked its performance on unextracted swab samples against an antigen test. Remarkably, despite a decrease in sensitivity (77%), direct ADESSO largely outperformed the antigen test, which correctly detected less than half of the positive samples resulting in a sensitivity of 46% (Fig. 2 and Table 1). Based on a Ct value distribution analysis across 6,439 infected individuals, we could estimate that using ADESSO as a POC test on unextracted samples could increase the portion of detected COVID-19 population by 31% in comparison to the antigen tests currently in use (Fig. 4). Here, we have also performed a side-by-side comparison of two sampling methods, namely NP and gargling in saline water. Our results show a general drop in sensitivity and LoD for saline gargle samples independently of the detection method used (Table 1 and Fig. 2c). While the results are in agreement with other studies[45,46], it is hard to discern the reasons behind this drop. One possibility is that gargling results in less cells and viral particles in the specimens in comparison to swabs.

This hypothesis would be in line with the observed higher RT-qPCR Ct values in gargle specimens compared to their matched swab samples (Supplementary Fig. 4e–g). Furthermore, our results show a disagreement between LoD on serial dilutions of synthetic viral genome and LoD in clinical samples. Despite the "synthetic" LoD of 2.5 cp/µl (~Ct 35; Fig. 1), the real clinical sensitivity of ADESSO corresponds to Ct 30–32 (Fig. 2) and the same is true for other studies although it has never been clearly articulated[28,30]. This aspect highlights that an extensive validation on real clinical samples covering the full range of viral titers, as the one shown here, is necessary to determine the real LoD of a diagnostic test. This is crucial to allow a fair comparison between sensitivities resulting from independent studies, which can be greatly influenced by the choice of the tested population.

Additionally, we demonstrated that ADESSO can be quickly adapted to recognize both single-nucleotide variants and deletions, and thus specifically identify the presence of a specific SARS-CoV-2 variant (Fig. 3). While another CRISPR-Dx method is able to detect SARS-CoV-2 Alpha, Beta and Gamma variants[56], ADESSO also enables sensitive detection of Delta and Omicron, increasing the number of recognizable variants using CRISPR-Dx technologies.

Finally, another important aspect of ADESSO is its affordability. We calculated a cost per reaction of 2.71€ and 4.88€ for fluorometric and lateral-flow detection, respectively (Supplementary Table 5), which is comparable to antigen tests and lower than any other detection method (Table 2). The cost would be even lower at a large-scale production. Altogether, ADESSO is cheaper than any RT-qPCR-based COVID-19 diagnostic test[57] and offers a more accessible option for widespread and more frequent testing. At the same time, ADESSO is comparable to the commonly used antigen tests in terms of cost[58], but offers a higher sensitivity.

Despite the higher sensitivity of direct ADESSO in comparison to other CRISPR-Dx technologies performed on unextracted samples[26,28,59], these other methods present the advantage to be one-pot reactions. This feature allows the reaction to run at one temperature, reducing handling and the risk of sample contamination. ADESSO is a two-step method and this certainly represents a limitation of our current study. Future research is necessary to generate a one-pot ADESSO that will retain high sensitivity as the one shown here. In this instance, testing other RT enzymes[60] and Cas13[22] proteins will be essential.

With the COVID-19 pandemic entering into its third year, it has become clear that time plays a critical role in the management of such an emergency. In order to control it, while waiting to

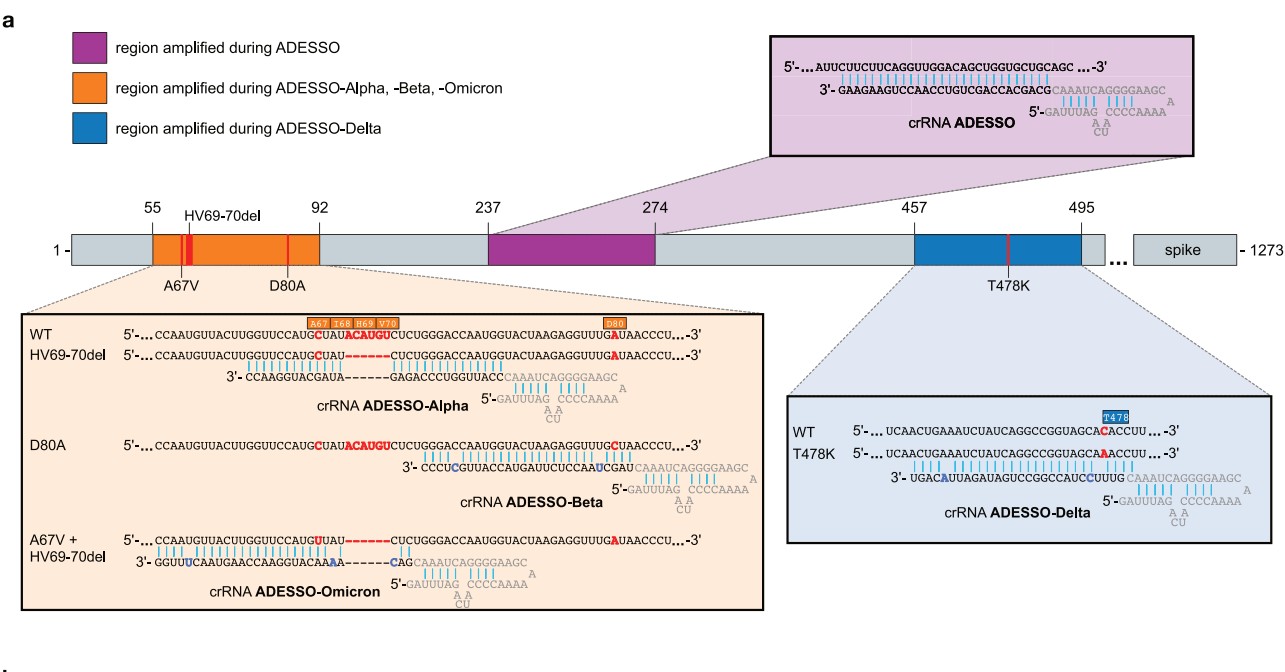

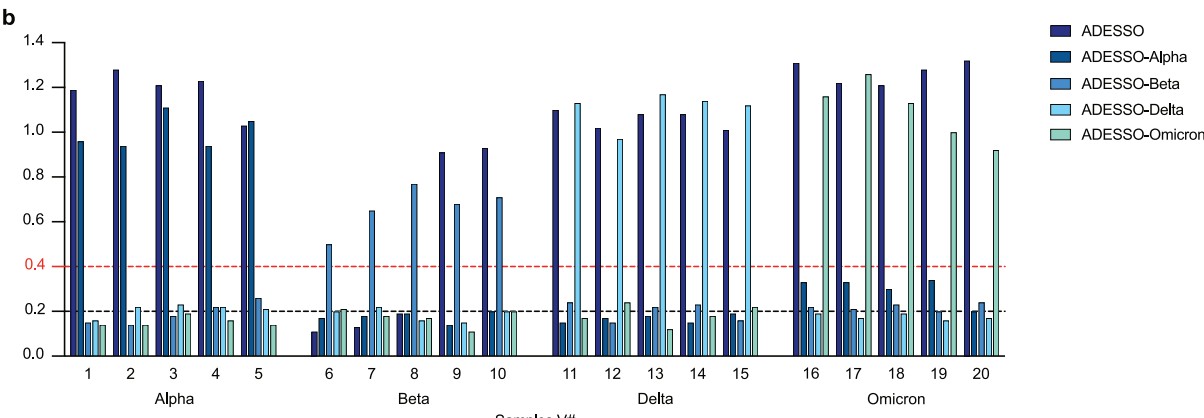

**Fig. 3 Adaptation of ADESSO for detection of SARS-CoV-2 variants. a** Schematic of SARS-CoV-2 S gene with annotation of the regions amplified during the different ADESSO tests (highlighted in different colors). The specific regions (and mutations) recognized by the different crRNAs are also shown for each test. Genomic sites that are mutated in at least one SARS-CoV-2 variant are highlighted in red. The synthetic mismatches introduced within the crRNAs are highlighted in blue. **b** SARS-CoV-2 variant detection by ADESSO, ADESSO-Alpha, ADESSO-Beta, ADESSO-Delta and ADESSO-Omicron in 20 clinical samples. Dashed lines represent the threshold to distinguish between positive and negative samples in ADESSO (black line) and ADESSO-variant (red line).

evaluate the long-term effect of the ongoing worldwide vaccination campaign, rapid detection of new infections and identification of variants are key factors. Here, we demonstrate that ADESSO could become a great tool in the fight against the virus. While restrictions are being lifted in many countries despite the not negligible infection rates, testing to keep the virus spread under control is still crucial, more than ever now.

## Methods

**Cas13 purification**. Plasmid encoding LwaCas13 (pC013 - Twinstrep-SUMO-huLwCas13a was a gift from Feng Zhang; Addgene plasmid #90097; http://n2t.net/addgene:90097; RRID: Addgene_90097)[25] was transformed into Rosetta cells and purified according to established protocols with substantial modification. Single colonies were pre-inoculated into 25 ml Luria Broth (LB) (100 μg/ml AMP) and grown overnight at 37 °C. This preinoculation was used to inoculate 4–12 L of Terrific Broth (TB) and let grown to an OD of 0.6 at 37 °C degrees while shaking at 150 rpm. The suspension was chilled for 30 min at 4 °C and subsequently induced with 0.5 mM IPTG and left shaking for an additional 16 h at 21 °C. Cells were harvested by centrifugation at 7,000 g for 30 min at 4 °C. The pellet was resuspended in 4× (wt/vol) supplemented lysis buffer (12 cOmplete Ultra EDTA-free tablets, 600 mg of lysozyme and 6 μl of benzonase to lysis buffer (20 mM Tris pH 8.0, 500 mM NaCl, 1 mM DTT)) and lysed by sonication. Lysate was cleared by

centrifugation at 10,000 g for 1 h at 4 °C. Cas13 was purified from the supernatant by nickel-affinity chromatography either using a 1 ml HIS-Trap column (Cytiva) and an ÄKTA pure FPLC system, or by Ni-NTA-agarose (Qiagen) gravity flow, with lysis buffer for the washing steps and a high-concentration imidazole elution. After initial purification, the protein sample was dialyzed overnight at 4 °C against lysis buffer to remove the imidazole and afterwards incubated with SUMO protease (ThermoScientific, 15 units/mg protein) at 4 °C overnight to remove the affinity tags. The sample was then re-applied to a 1 ml HIS-Trap column. Both the SUMO protease (which itself has a 6xHIS tag) and the cleaved affinity tag bound to the resin, while pure Cas13 eluted in the wash step using lysis buffer. A final size-exclusion chromatography step was performed using the ÄKTA pure system using 10 mM HEPES pH 7.0, 5 mM MgCl₂, 1 M NaCl and 2 mM DTT as gel filtration buffer on a Superdex 16/600 200 pg column. Purified protein was dialyzed against storage buffer (50 mM Tris-HCl pH 7.5, 600 mM NaCl, 5% glycerol, 2 mM DTT) for long term storage. The procedure and purification results are summarized in Supplementary Fig. 6.

**Synthetic SARS-CoV-2 RNA**. Fully synthetic wild-type (MT007544.1 or MN908947.3) and Delta (EPI_ISL_2695467) SARS-CoV-2 RNA was purchased from Twist Biosciences. In order to test SHERLOCK sensitivity, serial dilutions were prepared in water or in saline, from the initial concentration of 10⁶ cp/μl to 0.01 cp/μl.

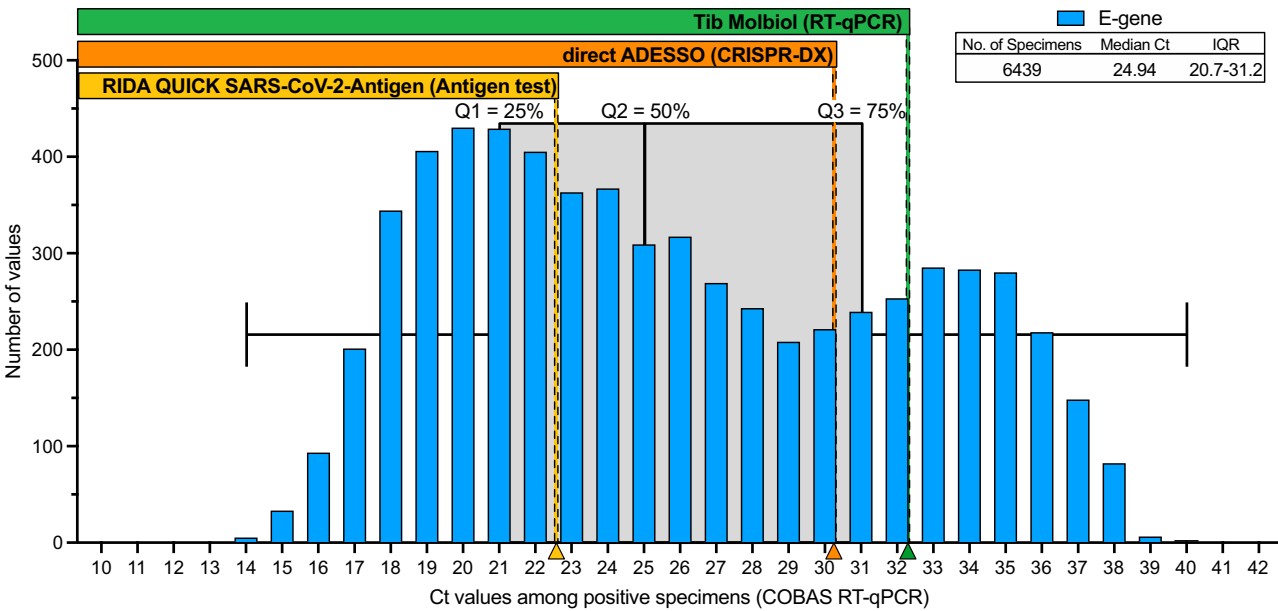

**Fig. 4 Ct value distribution across a population of non- or mildly symptomatic COVID-19 positive individuals.** The histogram shows the distribution of Ct values detected by using COBAS RT-qPCR (SARS-CoV-2 E gene) across 6,439 positive samples. These were collected from ambulatory patients showing minimal to mild symptoms. The box plot of these Ct values is represented in grey. Center line denotes the median (Ct = 24.94), bounds denote IQR (20.7–31.2) and whiskers denote minimum (13.73) and maximum (40.05). Quartiles (Q) 1, 2, 3 are also indicated to highlight the 25, 50, and 75% portions of the population, respectively. Arrowheads on the x axis represent the Ct values corresponding to the LoD of the three different detection methods, as evaluated in this work (Fig. 2). The rectangles above the histogram indicate the range of Ct values and the portions of specimens that would be detected by the three tests compared in this study (antigen test in yellow, direct ADESSO in orange and RT-qPCR in green).

**Table 2 Comparison of ADESSO assay with other widespread assays for the detection of SARS-CoV-2.**

|  | Antigen test | ADESSO | | Rapid RT-qPCR | RT-qPCR |
|---|---|---|---|---|---|
|  |  | Direct | on RNA |  |  |
| Clinical LoD (Ct value) | 22[16]–25[15] | 30 | 32 | 29[14]–35[15] | 36 |
| Assay reaction time | 5–20 min | 60 min | 60 min* | 45 min | 120 min* |
| RNA extraction | No | No | Yes | No | Yes |
| Sophisticated equipment needed | No | No | No | Yes | Yes |
| POC suitability | Yes | Yes | No | Yes | No |
| Cost per reaction | 4,80€[58] | 2–5€ | 2–5€* | 15–20€[57] | 9,11€[58] |

*without RNA extraction.

**In-vitro transcribed SARS-CoV-2 S and Orf1a RNA fragments**. These in-vitro transcribed (IVT) fragments were used only in the experiment shown in Supplementary Fig. 1c S1D. SARS-CoV-2 RNA, a kind gift of Prof. Bartenschlager (DKFZ, Heidelberg), was used for OneStep RT-PCR (Qiagen, #210212) as follows: 11 µl of nuclease-free water, 5 µl of 5× OneStep RT-PCR buffer, 1 µl of dNTP mix (10 mM each), 1.5 µl of each primer (forward and reverse, both 10 µM) and 1 µl of OneStep RT-PCR Enzyme Mix were added to 4 µl of denatured RNA. The primers used for the amplification of SARS-CoV-2 S gene and Orf1a gene are listed in Supplementary Table 6. The RT-PCR protocol was run as follows: retro-transcription at 50 °C for 30 min, denaturation at 95 °C for 15 min, followed by 40 cycles of denaturation at 94 °C for 30 sec, annealing at 61 °C (Orf1a gene) or 62 °C (S gene) for 30 sec and elongation at 72 °C for 5 sec. In the end a final elongation step at 72 °C was run for 10 min. PCR clean-up was performed on the RT-PCR products according to the manufacturer's instructions (Macherey-Nagel, #740609.250). The purified DNA was in-vitro-transcribed into RNA with the HiScribe T7 Quick High Yield RNA Synthesis Kit (NEB, #E2050S) following the suggested protocol for short transcripts. The IVT products were then treated with DNase I (HiScribe T7 Quick High Yield RNA Synthesis Kit, NEB, #E2050S) and purified with Monarch RNA Cleanup Kit (NEB, #T2050). The concentration of the purified products was determined with the Qubit RNA broad range (BR) kit (ThermoFisher Scientific, #Q10211). In order to test SHERLOCK sensitivity, serial dilutions were made in water from a concentration of 1 µM to 1 aM.

**Human clinical specimen collection and ethics statement**. Clinical specimens were collected at the Medical University Mannheim, Germany. The Ethics

Committee II of the University of Heidelberg (Medical Faculty of Mannheim) ruled the ethics for all the clinical samples used in this study. The committee reviewed and approved the proposal for the collection and use of NP and gargle samples (ref. 2020-556 N). Regarding the use of nose-throat samples, the ethic committee reviewed the specific proposal and concluded that according to the professional code of conduct for doctors and German regulations, the evaluation by an ethics committee was unnecessary. NP swabs and gargle samples were collected from ambulatory patients presenting minimal to mild symptoms or sent by the German Health Department after having contact with a SARS-CoV-2 positive person. After verbal and visual instruction gargling was performed with 8 ml of sterile 0.9% saline (Fa. Fresenius Kabi, Bad Homburg, Germany). Samples were collected in sterile containers without additives and stored at 4 °C until testing with PCR within 36 h. NP specimens were collected with flocked swabs (Improswab, Fa. Improve Medical Instruments, Guanzhou/China) and washed out with 2 ml 0.9% saline within 12 h of collection. For sample inclusion in the validation study and side-by-side comparison of ADESSO and RT-qPCR, initial PCR was performed on NP swab samples as part of routine clinical care using the cobas 6800 system (Roche, Penzberg, Germany) according to the manufacturer's instructions. Based on the results of the initial PCR, 95 positive and 100 negative samples were selected.

**RNA extraction**. For the first blind test (Supplementary Fig. 1), RNA was extracted from the clinical samples with the QIAamp® Viral RNA Mini kit (Qiagen, #52904) following the manufacturer's instructions (140 µl of swab were extracted and eluted in 60 µl). For the validation study (Fig. 2), RNA was extracted from 200 µl of the selected gargle and NP specimens with the MagnaPure Compact System (Roche,

Penzberg, Germany) using the Nucleic Acid isolation Kit I (Roche) resulting in 100 μl of eluate. The residual volume of gargle and NP specimens was stored at 4 °C and sent to the DKFZ for further analysis.

**RT-qPCR.** CDC taqman RT-qPCR initially (Supplementary Fig. 1) was performed in technical triplicates according to published protocols[61], which we adapted to a 384-well plate format and a reduced reaction volume of 12.5 μl. The reaction was performed using the Superscript III One-Step RT-PCR kit with Platinum Taq Polymerase. Magnesium sulfate and BSA were added to the reaction to a final concentration of 0.8 mM and 0.04 μg/μl, respectively. Primers and probes for the viral N1 and N2 and the human RNase P genes were added as ready-made mix (1 μl; Integrated DNA Technologies Belgium; CatNo. 10006713). The E-gene probes and primers (GATC, Germany) were used at final concentrations of 500 nM for each primer and 125 nM for the probe. ROX was added to a final concentration of 50 nM. PCR was performed in a QuantStudio 5 thermocycler, with cycling conditions 55 °C for 10 min, 95 °C for 3 min, followed by 45 cycles of 95 °C for 15 s and 58 °C for 30 s.

For the validation study (Fig. 2), real-time PCR of 10 μl RNA-eluate was performed on a BioRad CX96 cycler using the Sarbeco E-Gen-Kit (Fa. Tib Molbiol, Berlin, Germany) following the manufacturer's instructions. The residual volume of extracted RNA from gargle and NP specimens was stored at −20 °C and sent to the DKFZ for further analysis.

**Lysis of clinical samples for direct SARS-CoV-2 detection.** Clinical samples were lysed for direct ADESSO assay (Fig. 2) as follows: after vortexing, 10 μl of the sample were mixed with 10 μl of QuickExtract DNA Extraction solution (Lucigen, #QE09050) enriched with Murine RNase Inhibitor (NEB, #M0314) at a final concentration of 4 U/μl. Samples were then incubated at 95 °C for 5 min. After incubation, samples were mixed by vortexing and spun down for 15 seconds at 10,000 g. Finally, 5.6 μl of the sample (for RT-RPA 2X) were collected from the upper liquid phase, carefully avoiding to aspirate any precipitate, and used in the RT-RPA step.

**crRNA synthesis and purification.** All CRISPR-RNAs (crRNAs) used in this study are listed in Supplementary Table 6. To produce the crRNAs, we followed a previously published protocol[40]. In short, the templates for the crRNAs were ordered as DNA oligonucleotides from Sigma-Aldrich with an appended T7 promoter sequence (listed in Supplementary Table 6). These oligos were annealed with a T7-3G oligonucleotide and input in an in vitro transcription (IVT) reaction (HiScribe T7 Quick High Yield RNA Synthesis Kit, NEB, #E2050S). The crRNAs were then purified using Agencourt RNAClean XP Kit (Beckman Coulter, #A63987). The correct size of the crRNAs was confirmed on a Mini-PROTEAN TBE-Urea precast gel (Bio-Rad, #4566033) and the concentration evaluated by NanoDrop. Aliquots of each crRNA at the working concentration were produced to avoid repeated freeze and thaw cycles and stored at −80 °C.

**RT-RPA.** RT-RPA reactions were carried out with TwistAmp Basic (TwistDx, #TABAS03KIT) with the addition of M-MuLV Reverse Transcriptase (NEB, #M0253) and Murine RNase Inhibitor (NEB, #M0314). The optimized reactions were run at 42 °C for 45 minutes in a heat block. Here are the details for the optimized reaction (so called 2xRT-RPA): two lyophilized pellets TwistAmp Basic are used to prepare the following master mix for 5 reactions: 59 μl of Rehydration Buffer (RB) are mixed with 2.5 μl of each primer (forward and reverse) at a concentration of 20 μM, 1.5 μl of M-MuLV RetroTranscriptase (200 U/μl - NEB, #M0253) and 1.5 μl of RNase Inhibitor, Murine (40 U/μl - NEB, #M0314). The RB-primer-enzyme mix is used to rehydrate two pellets and finally 5 μl of MgOAc are added. The complete mix is aliquoted (14.4 μl) on top of 5.6 μl of each sample. The RT-RPA protocol was optimized throughout the study. To avoid any confusion, we provide detailed protocols for each experiment presented in this work in Supplementary Methods. Primer sequences are provided in Supplementary Table 6.

**Cas13 cleavage reaction for lateral flow readout.** The optimized reaction mix for Cas13 activity was prepared by combining 4.3 μl of nuclease-free water, 1 μl of cleavage buffer (400 mM Tris pH 7.4), 1 μl of LwaCas13a protein diluted in Storage Buffer (SB)[40] to a concentration of 126.6 μg/ml, 0.5 μl of crRNA (40 ng/μl), 0.5 μl of lateral flow reporter (IDT, diluted in water to 20 μM), 0.5 μl of SUPERase-In RNase inhibitor (ThermoFisher Scientific, #AM2694), 0.4 μl of rNTP solution mix (25 mM each, NEB, #N0466), 0.3 μl of NxGen T7 RNA Polymerase (Lucigen, #30223-2) and 0.5 μl of MgCl₂ (120 mM). 1 μl of the RT-RPA-amplified product was then added to the mix and, after vortexing and spinning down, the mixture was incubated for 10 minutes at 37 °C in a heat block. The Cas13 protocol was optimized throughout the study. To avoid any confusion, we provide detailed protocols for each experiment presented in this work in Supplementary Methods. The reporter sequence is provided in Supplementary Table 6.

**Lateral flow readout.** Lateral flow detection was performed using commercially available detection strips (HybriDetect — Universal Lateral Flow Assay Kit, Milenia Biotec GmbH, Gießen, #MGHD 1). The 10μl-LwaCas13a reactions were

transferred to a tube already containing 80 μl of HybriDetect Assay buffer. After vortexing and spinning down the reaction mix, a lateral flow dipstick was added to the reaction tube. The result was clearly readable after one minute. Once the whole reaction volume was absorbed, the dipstick was removed and photographed with a smartphone camera for band intensity quantification performed with the freely available ImageJ image processing program[62]. The results are shown as intensity ratio (test band/control band) and tests were considered positive for values of intensity ratio above 0.2 based on the results shown in Supplementary Fig. 2.

**Cas13 cleavage reaction for fluorescence readout.** The reaction mix for Cas13 activity was prepared by combining 8.6 μl of nuclease-free water, 2 μl of cleavage buffer (400 mM Tris pH 7.4), 2 μl of LwaCas13a protein diluted in Storage Buffer (SB) to a concentration of 126.6 μg/ml, 1 μl of crRNA (40 ng/μl), 1 μl of the fluorescent reporter (IDT, diluted in water to a final concentration of 4 μM), 1 μl of Murine RNase inhibitor (NEB, #M0314), 0.8 μl of rNTP solution mix (25 mM each, NEB, #N0466), 0.6 μl of NxGen T7 RNA Polymerase (Lucigen, #30223-2) and 1 μl of MgCl₂ (120 mM). 2 μl of the RT-RPA-amplified product were then added to the mix. The 20μl-LwaCas13a reactions were transferred in 5μl-replicates (4 wells each sample) to a 384-well, round, black-well, clear-bottom plate (Corning, #3544). The plate was briefly spun down at 500 g for 15 sec to remove potential bubbles and placed into a preheated GloMax® Explorer plate reader (Promega) at 37 °C. Fluorescence was measured every 5 min for 3 h. Data analysis, if not otherwise stated, was performed at the 30-min time-point. The reporter sequence is provided in Supplementary Table 6.

**RNAse activity detection assay.** In order to check for RNase activity in clinical samples, 10 μl of a negative sample, both as swab and gargle water, were mixed with 10 μl of QuickExtract DNA Extraction Solution with or without RNase Inhibitor, Murine (NEB, #M0314) at a final concentration of 4 U/μl. The samples were then incubated at 95 °C for 5 min. After incubation, RNaseAlert substrate v2 (RNaseAlert Lab Test Kit v2, Thermo Fisher Scientific, #4479768) was added at a final concentration of 200 nM. The samples were mixed by vortexing, spun down and incubated at RT for 30 min in the dark. After incubation, the samples were transferred to a 384-well, round, black-well, clear-bottom plate (Corning, #3544) in 5μl-replicates (4 wells each sample). The plate was briefly spun down at 500 g for 15 s to remove potential bubbles and placed into a GloMax Explorer plate reader (Promega). RNaseAlert substrate fluorescence was measured every 5 min for 30 min. Data analysis, if not differently stated, was performed at the 5-min time-point.

**Antigen test.** For the validation study (Fig. 2), RIDA QUICK SARS-CoV-2 Antigen test (R-Biopharm AG, #N6803) was performed following the manufacturer's instructions[44].

**Bioinformatic analysis of SARS-CoV-2 genomes.** All the SARS-CoV-2 genomes sequences were downloaded from GISAID on November 22, 2021. Fasta files were filtered using the pipeline for Augur[63] (Preparing your data — SARS-CoV-2 Workflow documentation). Only complete and high coverage genomes from humans were considered for further analysis. Frequency analysis of viral genetic variations was performed using Nextclade tool with standard parameters (Nextclade CLI). Sequences of primers and crRNA used in ADESSO were analyzed using a custom perl script. Percentages of exact binding are shown in Supplementary Fig. 5e.

**Statistical analysis and data panels generation.** All the statistical analysis and the data panels in this article were generated using Prism 8 (GraphPad). Statistical details for each experiment can be found in the figure legends.

**Reporting summary.** Further information on research design is available in the Nature Research Reporting Summary linked to this article.

## Data availability

**Primer and crRNA design.** Primers for RT-RPA and crRNAs for Cas13 detection were designed following the guidelines published for the SHERLOCK method[40] using NCBI Primer-BLAST[64], Primer3Plus[65] or ADAPT[66]. Specific information about each primer and crRNA is provided in Supplementary Table 6.

**Human clinical specimen.** Information regarding all the samples used in this study is available in Supplementary Tables 1–4. Sequencing information reporting mutations within the S gene for clinical samples carrying SARS-CoV-2 variants is available in Supplementary Data 1.

**Protocols.** The RT-RPA and Cas13 reaction protocols used for each experiment are provided in Supplementary Methods with reference to the corresponding figures. The exact volumes are given for one single reaction.

**Reagents and materials.** Detailed information about reagents and material used in this study is provided in Supplementary Table 7.

**Bioinformatic analysis database and output files.** All the SARS-CoV-2 genomes sequences analyzed here were downloaded from GISAID[67]. All the output files are available (Supplementary Data 2–4).

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

## Acknowledgements

We thank the transport service staff, German Cancer Research Center (DKFZ), for providing excellent transportation services of all the clinical samples from the Medical University Mannheim to the DKFZ. We thank the EMBL sequencing facility for providing excellent sequencing service of the genome of SARS-CoV-2 variants. The work was supported by a grant from the Ministry of Science, Research and the Arts of Baden-Württemberg for COVID-19 research (grant agreement no. Kap. 1499 TG 93 to Dr. Riccardo Pecori (DKFZ), Prof. Dr. Nina Papavasiliou (DKFZ) and Prof. Dr. Thomas Miethke (Medical Faculty of Mannheim)). Several schematics presented here were created with Biorender.com (Figs. 1f, 2a and Supplementary Figs. 1a, 3c).

## Author contributions

B.C. and R.P. designed the experiments. J.P.V., A.H., and C.E.S. produced Cas13 protein. B.C. and R.P. performed all the experiments using SHERLOCK/ADESSO. P.B., K.H.M., M.S., P.A.G., and B.R. optimized and performed confirmative RT-qPCR on clinical samples using CDC protocol. MK collected the specimens and A.G.K. performed the RT-qPCR on clinical samples using Tib Molbiol under the supervision of S.W. B.M. analyzed the RT-qPCR data. S.A. quantified the bands of the lateral flow strips. S.D.G. performed the bioinformatic analysis. B.C. and R.P. analyzed the data and wrote the manuscript. P.A.G. and J.P.V. contributed in editing and conceiving the structure of the final manuscript. R.P., T.M., and F.N.P. conceived the study and supervised the research. All authors have read and approved the manuscript.

## Funding

## Competing interests

RP, FNP and BC are inventors on the pending patent applications EP 20 173 912.5, which covers the ADESSO technology for SARS-CoV-2 detection, held by the DKFZ. All other authors declare no competing interests.
