## [Peer Review File · Nature Communications]

Rapid, adaptable and sensitive Cas13-based COVID-19 diagnostics using ADESSOREVIEWER COMMENTS

Reviewer #1 (Remarks to the Author):

Casati et al. presents improvement of virus diagnosis with CRISPR-Cas13 with a procedure named ADESSO. They first removed the step of RNA extraction, although this has been worked out by other in methods such as HUDSON and SHINE, and in reference 42. However, the authors were able to maintain high sensitivity that others were not able to achieve, although reference 42 came very close. They also applied ADESSO to distinguish two deletion variants, alpha and beta strains of SARS-CoV-2, from the wild-type.

The first optimization was at the step of RNA extract and RT-RPA steps. Authors used several commercially available RT enzymes (that were recombinantly made to lack a strong RNase H activity) with or without an independent RNase H. The results showed that the M-MuLV is the best without RNase H. They next further optimized the concentration of the commercially available RPA components and the recombinant Cas13 enzyme. The LoD was determined using either the lateral flow stipe or fluorescence readouts.

Authors then applied ADESSO on a large collection of clinical samples in parallel with two standard testing methods, RT-qPCR and antigen tests. They also characterized ADESSO with either extracted RNA or directly in order to compare with the antigen test. This large-scale test on 195 samples is useful and comprehensive.

Though the strength of the work is the access to large clinical samples and the parallel testing with three different methods and on two specimen collection methods, the weakness is the lack of innovation. The optimization strategies are quite similar to those outlined in references 42 and 44. It would have been higher impact if ADESSO can distinguish variants such as Delta from the wild-type. Both alpha and delta contain substantial difference in nucleotide sequence (6 and 9 nucleotide deletions, respectively), their pairing to Cas13 guide RNA would thus have been easily discriminated against. Authors simply realized this experimentally. It would be more challenging to distinguish variants with single nucleotide changes such as the Delta strain.

Major concerns.

1. The result on sample 11 raises some concerns. Figure 3 CD correctly shows that crRNA DHV crRNA can detect alpha variants (#1-#10) but not wild-type or Beta strain and the HV crRNA can detect Beta strain. When using the normal crRNA in ADESSO, sample 11 was not detected (note that samples #12 and #13 are detected). Authors argued that this is because of a single additional mismatch in #11 that failed to amplify in the RT-RPA step. Can authors clarify where is the RPA primer in normal ADESSO? In addition, does ADESSO use a different crRNA than the HV crRNA? It appears to be that shown Figure S5D (Supplemental file #3). If so, Samples 12 or 13 should not work either due to a 9-nucleotide mismatch. But if authors did get product using this RPA primer on the Beta strain (Figure S5D), it would suggest that the RT is quite non-specific.

If ADESSO has a different crRNA, it would be beneficial to draw its secondary structure as those for HV crRNA in Figure 3B to avoid the confusion.

Related: Figure S5D, sample 11, the last A in RPA primer should be paired, thus it only has a AG bulge;

Even if authors can clarify the various crRNA and RPA primers used, it unlikely changes the fact that this method as it stands will unable to distinguish the most dominant variants that are single-base mutations (within the complementary region);

2. Figure S3A seems to show that sample 11 is detectable. Which guide RNA was used here? Furthermore, was a 4x RPA concentration an outlier?

3. Given the concern that ADESSO is not really capable of detect most variants, the statements referring to this capability should be revised or removed. For instance, the last sentence in the abstract "...quickly adjusted to detect new variants."; Lines 290 in Discussion, "ADESSO can be

adapted to any variant of concern..." is not valid at all;

4. It is not clear how in vitro transcribed RNA copy numbers were obtained. UV absorbance would be inaccurate. This quantification is important to the claim of sensitivity;

5. Figure 2 BC right vertical axis looks like Ct value just like the left vertical axis. They would better be stripe intensity ratios. In addition, to account for antigen test results, perhaps an independent axis for the antigen assay readout;

6. It is probably worth mentioning that the commercially available RT enzymes have weak or no RNase H activities. Thus, independent RNase H was used to optimize ADESSO (Figure 1);

7. For fluorescence detection, no background fluorescence was measured or corrected. It appears that authors used 4 microMolar probe that could have substantial background intensity;

8. There is no description of statistical treatment of the technical replicates;

9. Do authors have an explanation why Ct values have a bimodal distribution in Figure 4?

Reviewer #2 (Remarks to the Author):

The manuscript entitled "ADESSO: a rapid, adaptable and sensitive Cas13-based COVID-19 diagnostic platform" introduced a new point-of-care approach for COVID-19 diagnosis, including variant identification. The author validated this platform in the large size of the clinical cohort and systematically compared it with current RT-qPCR and antigen POC tests. Their data is solid and well organized, however, it lacks fundamental novelty as a new diagnosis method.

Major comments

1. The author claimed, "evaluated alternative reagents and different reaction conditions for RNA extraction", but besides reverse transcriptase, which reagents and conditions were optimized are not clear. Furthermore, the discussion of different enzymes is missing. Dr. Feng Zhang's lab has opened their detailed protocol of direct SHERLOCK COVID-19 detection. In this protocol, Dr. Zhang's team has optimized isothermal amplification buffer, guide RNA design, salt and PH, even a new version of Cas13 to adapt one-step RPA-CRISPR. The author should provide optimization details to show which step/reagents/conditions are critical to elevating SHERLOCK sensitivity and specificity.

2. Compared to the Nature BME article (<https://doi.org/10.1038/s41551-020-00603-x>), I did not find much improvement in the methodological side. The gargle of saline lysate might be convenient for POC tests, but uncleared multiple-step operations (lysis, RPA, CRISPR) may reduce the feasibility for users without training. One example is the volume of oral wash will not be equal among individuals, then how much of lysis buffer should be added to the sample? The incubation temperature is another concern, 95 °C is difficult for POC tests without lab equipment.

3. It is good to see ADESSO-Alpha assay showed the ability to detect Alpha or Beta strain. However, the selected guide RNA can only distinguish these two strains. What if we apply ADESSO-Alpha to other variants, Delta strain, for example? Moreover, if the sample is from an unknown infection, how could we tell the results?

Minor comments

1. What kind of lysis buffer is used for antigen POC test? QuickExtract DNA Extraction solution is not suitable for protein analysis.

2. RT-qPCR can reach 1 copy per μL per CDC EUA, not 2.5

3. Line 101 "show a lower limit of detection (LoD)" should be "a higher limit of detection (LoD)" or "lower sensitivity".

Reviewer #3 (Remarks to the Author):

Casati and colleagues present and validate their method, ADESSO, which is an optimized SHERLOCK-based method for detection of SARS-CoV-2 RNA and specific Alpha and Beta variants. ADESSO can use both extracted or non-extracted nasal and saliva samples as input and has a paper-based lateral flow readout for the ultimate goal of being used in point-of-care (POC) settings. The authors extensively optimize their SARS-CoV-2 assay, which like many SHERLOCK assays uses RT-RPA for amplification and Cas13a for detection, and thoroughly validate their optimized assay on hundreds of patient samples comparing ADESSO's results to RT-qPCR and a rapid antigen test. The authors then demonstrate that this platform can quickly be updated (~1 week) to test for specific variants, a quite valuable feature given the rise of more transmissible variants and/or those that respond differentially to therapies and vaccines.

I first want to applaud the authors for the following strengths of this manuscript:

1. This manuscript is very well written and beautifully organized. The descriptions of the results are clear and easy to follow, overall making this manuscript a joy to read.
2. The inclusion of upstream sample processing methods that can be performed quickly and without extensive hands-on time (i.e. the inclusion of QuickExtract) is an appreciated component of this work particularly for the development of POC tests.
3. The completion of both extensive testing (hundreds of samples) to compare the performance of ADESSO against both RT-qPCR and antigen tests as well as thorough testing of negative samples to set a proper and experimentally validated threshold for calling positive and negative samples, a challenging aspect for visual readouts.

I have outlined below some major and minor suggestions to improve the manuscript:

Major:

1. I applaud the authors for including references to a majority of the relevant, previously published CRISPR-based SARS-CoV-2 diagnostics in their introduction, but there are a couple concepts that lack sufficient discussion:
 - a. The authors are correct in stating in Lines 99-101, that many previously developed assays use extracted RNA or have lesser performance on non-extracted samples, but these referenced works use a single reaction set up where amplification and detection are combined in one tube. Although this could be one of the reasons the sensitivity of these assays is reduced, having a single reaction is huge advantage because (1) the reaction operates at one temperature, (2) less manipulations are required which saves reaction set-up time and assay performance time, and (3) reduces the number of times an amplified sample is manipulated which increases the risk of sample contamination and false positives.
 - b. The authors should discuss their approach in the context of the previously published miSHERLOCK technology (doi: 10.1126/sciadv.abh2944) which applies a one-pot SHERLOCK assay (RPA + Cas12) to detect SARS-CoV-2 variants (both deletions and single-nucleotide changes) using a device that completely automates the assay from sample collection to smartphone-reported results. I think the addition of a Delta variant assay would better differentiate this manuscript from miSHERLOCK and provide an example of a single-nucleotide variant ADESSO assay.
2. The authors should perform some additional technical validation of the saliva-based assay and with the antigen test which could improve or explain some of the differences in sensitivity with this sample type:
 - a. Figure 1A: Does the QuickExtract method with saliva as input have reduced RNase activity at a similar level as observed with the nasal sample? If not, this could explain the reduced ADESSO LOD for saliva in Figure 1C including some samples with low Ct values and suggest the need for additional units of RNase when testing this sample type.

b. Figure 2C: After quick glance at the product website, it is unclear whether or not the antigen-based test used is validated and compatible with saliva samples. This could explain how only 5 samples were positive by this test and many samples including those with low Ct values failed to test positive. This could potentially be tested by adding SARS-CoV-2 synthetic RNA into a solution with increasing amounts of healthy saliva to see if saliva has an inhibitory effect on the antigen test. I request this instead of retesting patient samples because I know that often patient sample volume is limited.

3. In some instances, there needs to be a bit more clarity regarding ADESSO and direct ADESSO's LOD and sensitivity when compared to other assays:

a. In Figure 1B-D: Clarify whether these assay optimizations and LOD experiments were performed with 2.5 uL of just synthetic RNA or RNA spiked into a mixture of RNase inhibitor, QuickExtract, and a healthy swab sample. It is possible that Quick Extract could impact the performance of both RPA and detection and therefore the sample processing mix should be considered in the context of sample input volume and the LOD of the ADESSO Direct assays.

b. Figure 2B-C: The authors are underselling the performance of their assay compared to RT-qPCR, because their main comparator are the results of the COBAS RT-qPCR which was performed previously and not on the same RNA extract used for ADESSO testing. This could particularly affect samples with high Ct values which could be more susceptible to freeze-thaw and sample extraction causing variation in the RNA concentration. The side-by-side testing against Tib Molbiol would be a more accurate comparison.

c. In Figure S5: the authors show that a few of their samples had variation at the RPA binding site in their ADESSO assay. Is it possible that some of this variation could have led to some false negatives within the Figure 2B-C data? The authors should discuss this, include text to describe how their primers and crRNA were designed, and evaluate how much diversity exists within their designed assay amplicon compared to available sequence data.

Minor:

1. Including line numbers makes it easier to reference specific lines in the manuscript and adding them in myself could mean some of the line number references are off.

2. Figure 1B: Because neither the RT enzyme amount nor sample input were held constant in the results displayed, it is unclear if additional RT enzyme input with 2.5 uL of sample could have an added benefit on the sensitivity of the assay. It is also worth noting that increased volume of sample inherently should increase assay sensitivity because additional copies are added into the reaction.

3. Figure 1C: I encourage the authors to take a look at [doi:10.1038/s41467-020-19258](https://doi.org/10.1038/s41467-020-19258) where the authors test many RT enzymes +/- RNase H with RT-RPA. The results are quite similar to ones presented here, but a few of the enzymes (e.g. SSIV or Maxima with the inclusion of RNase H) could improve ADESSO's performance.

4. Figure S3A and C: Missing T and C labels.

5. Lines 232-236: Clarify whether there was any confirmatory assay to determine which samples were Alpha versus Beta.

6. Line 273: ADESSO -> ADESSO Direct

7. Lines 273-274: It might be more compelling to discuss the sensitivity by sample type (NP versus saliva) per my comments above.

8. Lines 290-305: I would consider incorporating this into the results section and not the discussion.

Response to referees letter

April 4th, 2022

Dear Reviewers,

Thank you for reviewing our manuscript entitled, "**ADESSO: a rapid, adaptable and sensitive Cas13-based COVID-19 diagnostic platform**", for publication in *Nature Communications*. We are grateful for the insightful comments and critiques and have addressed them below in a point-by-point manner.

We look forward to your decision.

With best regards,

Dr. Riccardo Pecori,
(for the authors)

Response to Reviewers comments

Reviewer #1 (Remarks to the Author):

Casati et al. presents improvement of virus diagnosis with CRISPR-Cas13 with a procedure named ADESSO. They first removed the step of RNA extraction, although this has been worked out by other in methods such as HUDSON and SHINE, and in reference 42. However, the authors were able to maintain high sensitivity that others were not able to achieve, although reference 42 came very close. They also applied ADESSO to distinguish two deletion variants, alpha and beta strains of SARS-CoV-2, from the wild-type.

The first optimization was at the step of RNA extract and RT-RPA steps. Authors used several commercially available RT enzymes (that were recombinantly made to lack a strong RNase H activity) with or without an independent RNase H. The results showed that the M-MuLV is the best without RNase H. They next further optimized the concentration of the commercially available RPA components and the recombinant Cas13 enzyme. The LoD was determined using either the lateral flow stipe or fluorescence readouts.

Authors then applied ADESSO on a large collection of clinical samples in parallel with two standard testing methods, RT-qPCR and antigen tests. They also characterized ADESSO with either extracted RNA or directly in order to compare with the antigen test. This large-scale test on 195 samples is useful and comprehensive.

Though the strength of the work is the access to large clinical samples and the parallel testing with three different methods and on two specimen collection methods, the weakness is the lack of innovation. The optimization strategies are quite similar to those outlined in references 42 and 44. It would have been higher impact if ADESSO can distinguish variants such as Delta from the wild-type. Both alpha and delta contain substantial difference in nucleotide sequence (6 and 9 nucleotide deletions, respectively), their pairing to Cas13 guide RNA would thus have been easily discriminated against. Authors simply realized this experimentally. It would be more challenging to distinguish variants with single nucleotide changes such as the Delta strain.

Response: We thank the reviewer for the cogent summary and the insightful suggestions. We have now included new data for specific detection of Beta, Delta, and Omicron variant samples based on single nucleotide changes (new Figure 3, S5). We are confident that these new data provide additional novelty to our work.

Major concerns.

1. The result on sample 11 raises some concerns. Figure 3 CD correctly shows that crRNA DHV crRNA can detect alpha variants (#1-#10) but not wild-type or Beta strain and the HV crRNA can detect Beta strain. When using the normal crRNA in ADESSO, sample 11 was not detected (note that samples #12 and #13 are detected). Authors argued that this is because of a single additional mismatch in #11 that failed to amplify in the RT-RPA step. Can authors clarify where is the RPA primer in normal ADESSO? In addition, does ADESSO use a different crRNA than the HV crRNA? It appears to be that shown Figure S5D (Supplemental file #3). If so, Samples 12 or 13 should not work either due to a 9-nucleotide mismatch. But if authors did get product using this RPA primer on the Beta strain (Figure S5D), it would suggest that the RT is quite non-specific.

Response: We thank the reviewer for highlighting this problem. Normal ADESSO recognizes the region in purple in new Figure 3a. The region recognized by ADESSO-Alpha is different, as highlighted in orange in new Figure 3a. Therefore, also the two crRNAs specific for these two regions are different. We have now

added a new panel (now panel d) in Figure S5 and generated a new figure 3 to allow a better understanding of the different ADESSO tests.

We were also surprised to observe that RT-RPA works in the presence of a 9 nt deletion for samples 12-13 (now samples v9 and v10). This is probably because primer annealing during RPA is mediated by recombinases. Thus, the only difference between sample 11 (now sample v6) and samples 12-13 (now samples v9 and v10) is the point mutation leading to R246I. There aren't any other differences either within the sequence recognized by the assay or in Ct values between samples (old Figure S5C-D and Supplementary File 2.1). We have now included in the analysis two additional Beta samples (v7,8) with the same mutagenesis pattern as v6 and we could reproduce the same results, thus demonstrating that ADESSO test results negative only when Beta samples have both Δ 242-244 and R246I (see new Figure 3 and S5).

Finally, we agree with the reviewer that RPA alone is not specific. This aspect seems to be shared between all isothermal amplification methods (e.g., LAMP, as stated in ref. 42). However, ADESSO as other CRISPR-DX methods takes advantage of the specificity of the Cas13 protein, mediated by the crRNA, so that any potential unspecific product obtained during isothermal pre-amplification won't be detected.

If ADESSO has a different crRNA, it would be beneficial to draw its secondary structure as those for HV crRNA in Figure 3B to avoid the confusion.

Response: We have now modified Figure 3 accordingly. We are confident that the new figure is explicit enough to avoid any confusion.

Related: Figure S5D, sample 11, the last A in RPA primer should be paired, thus it only has a AG bulge;

Response: We thank the reviewer for noting this. We have now modified Figure S5d accordingly. Please note that the samples names have changed.

Even if authors can clarify the various crRNA and RPA primers used, it unlikely changes the fact that this method as it stands will be unable to distinguish the most dominant variants that are single-base mutations (within the complementary region);

Response: We now provide data where specific detection of Beta, Delta and Omicron variant samples is based on the discrimination of single-base mutations (see new Figure 3, S5). We believe these new data are sufficient to prove the ability of ADESSO to discriminate single point mutations and thus SARS-CoV-2 variants.

2. Figure S3A seems to show that sample 11 is detectable. Which guide RNA was used here? Furthermore, was a 4x RPA concentration an outlier?

Response: We thank the reviewer for pointing out this source of misunderstanding. Sample 11 in Figure S3 is the one used in the experiment shown in Figure S1, and not the sample 11 from Figure 3. We have now updated the legend of Figure S3 and additionally renamed the samples in Figure 3 as "v#" (variants#) to clarify this point. Additionally, all the information about the clinical samples included in the study are available in Supplementary Tables 1-4. In these tables we also have indicated the figures in which each sample was used.

Yes, we have considered the 4x an outlier. Already 2x and 3x resulted in an improved reaction, thus we decided to proceed with 2x to maintain a lower cost per reaction. Additionally these results are in agreement with ref 47, so we decided not to repeat this experiment.

3. Given the concern that ADESSO is not really capable of detect most variants, the statements referring to this capability should be revised or removed. For instance, the last sentence in the abstract "...quickly

adjusted to detect new variants."; Lines 290 in Discussion, "ADESSO can be adapted to any variant of concern..." is not valid at all;

Response: We now provide data showing specific detection of Beta, Delta and Omicron variant samples based on detection of a single-base mutation. We believe these new data are sufficient to validate our statements.

4. It is not clear how in vitro transcribed RNA copy numbers were obtained. UV absorbance would be inaccurate. This quantification is important to the claim of sensitivity;

Response: We thank the reviewer for pointing this out. IVT RNA was used only in the experiment shown in new Figure S1c. The RNA was quantified via Qubit RNA broad range (BR) kit as now we have stated in the methods (line 392-393). Any other experiment uses synthetic SARS-CoV-2 RNA purchased from Twist Biosciences.

5. Figure 2 BC right vertical axis looks like Ct value just like the left vertical axis. They would better be stripe intensity ratios. In addition, to account for antigen test results, perhaps an independent axis for the antigen assay readout;

Response: We agree with the reviewer that the two axes are showing Ct values and they are repetitive. Our goal in making this figure was to show the difference in LoD (expressed as Ct value) between the different detection methods used. We have now updated a clearer version of this figure. All the band intensity ratios are shown in Figure S4.

6. It is probably worth mentioning that the commercially available RT enzymes have weak or no RNase H activities. Thus, independent RNase H was used to optimize ADESSO (Figure 1);

Response: We thank the reviewer for this comment. We have now modified the text accordingly (lines 157-159).

7. For fluorescence detection, no background fluorescence was measured or corrected. It appears that authors used 4 microMolar probe that could have substantial background intensity;

Response: The final concentration of the RNA reporter (probe) is 0.2 μ M. We have now updated Supplementary Methods (see new Supplementary Information) to make this clear. We have also updated Figure S3 to show the background-subtracted fluorescence. The new figure confirms the previous conclusions.

8. There is no description of statistical treatment of the technical replicates;

Response: We have now updated all the figures and figures legend to clarify this aspect.

9. Do authors have an explanation why Ct values have a bimodal distribution in Figure 4?

Response: This is a very interesting question. No, unfortunately, we don't have a real explanation for this observation.

Reviewer #2 (Remarks to the Author):

The manuscript entitled "ADESSO: a rapid, adaptable and sensitive Cas13-based COVID-19 diagnostic platform" introduced a new point-of-care approach for COVID-19 diagnosis, including variant identification. The author validated this platform in the large size of the clinical cohort and systematically compared it with current RT-qPCR and antigen POC tests. Their data is solid and well organized, however, it lacks fundamental novelty as a new diagnosis method.

Major comments

1. The author claimed, "evaluated alternative reagents and different reaction conditions for RNA extraction", but besides reverse transcriptase, which reagents and conditions were optimized are not clear. Furthermore, the discussion of different enzymes is missing. Dr. Feng Zhang's lab has opened their detailed protocol of direct SHERLOCK COVID-19 detection. In this protocol, Dr. Zhang's team has optimized isothermal amplification buffer, guide RNA design, salt and PH, even a new version of Cas13 to adapt one-step RPA-CRISPR. The author should provide optimization details to show which step/reagents/conditions are critical to elevating SHERLOCK sensitivity and specificity.

Response: We thank the reviewer for this comment. During the development of ADESSO, we have optimized each of the 3 steps in SHERLOCK, namely RNA extraction, RT-RPA amplification, and Cas13 detection. For RNA extraction we implemented the addition of an RNase inhibitor and we evaluated its effect both on swab and gargle samples (Fig. 1a). RT-RPA was optimized for input volume, amount of RT enzyme needed in a single reaction (Fig. 1b), and different RTs (plus/minus RNase H) were evaluated (Fig. 1c). Additionally, we have tested different amounts of RPA reagents (1X to 5X) (Fig. S3a,b). Cas13 detection was optimized for amounts of Cas13 and crRNA, resulting in a faster reaction (Fig. S3c,d). We have also experimentally defined the minimal time needed for a clear positive output on a lateral-flow stick (Fig. S3e,f), which allowed us to increase the RT-RPA time to 45 min (Fig. 1f), for highly sensitive reactions (ref. 57 - Kellner et al 2019). Everything was done keeping in mind the total cost per reaction that was kept as low as possible (summarized in discussion and described in detail in Suppl. Information). For the sake of clarity, all the protocols used in this work are available in Supplementary Information, under Supplementary Methods, as well.

Finally, we have also included a discussion of different enzymes in a completely rewritten new discussion section (lines 335-336).

2. Compared to the Nature BME article ([BLOCKEDdoi\[.\]org/10\[.\]1038/s41551-020-00603-xBLOCKED](https://doi.org/10.1038/s41551-020-00603-x)), I did not find much improvement in the methodological side. The gargle of saline lysate might be convenient for POC tests, but uncleared multiple-step operations (lysis, RPA, CRISPR) may reduce the feasibility for users without training. One example is the volume of oral wash will not be equal among individuals, then how much of lysis buffer should be added to the sample? The incubation temperature is another concern, 95 °C is difficult for POC tests without lab equipment.

Response: We agree with the reviewer that gargling of saline water would have been convenient for a POC test. Indeed, this is exactly why we included those samples in our extensive three-way comparison. Unfortunately, our results show that this sampling method leads to a drop in sensitivity for all the detection methods used here, as shown by others (ref 62). We speculate that the reason behind this effect is the high variability in performing the gargling between different people.

Regarding the technical questions, independently of the volume of oral wash for each individual, ADESSO always uses a volume of 1:1 between sample and lysis buffer (usually 10µl:10µl; see protocol details in Supplementary Methods). Finally, 95°C can be reached using a small thermoblock that could be used in a POC station. In countries with limited resources a hot plate to boil water would be enough to ensure the inactivation of the virus and the release of RNA. Little differences in temperature for the lysis

step do not represent a problem for the final detection (see e.g. Smyrlaki et al 2020, Nat Comm, PMID: 32968075).

3. It is good to see ADESSO-Alpha assay showed the ability to detect Alpha or Beta strain. However, the selected guide RNA can only distinguish these two strains. What if we apply ADESSO-Alpha to other variants, Delta strain, for example? Moreover, if the sample is from an unknown infection, how could we tell the results?

Response: We thank the reviewer for this comment. We now provide data also for specific detection of Beta, Delta, and Omicron variant samples based on the detection of single-base mutations. As for the second point, the reviewer is completely correct: while ADESSO, as any other detection method, cannot be programmed to detect an unknown sequence, it can be designed to recognize a very conserved region of the SARS-CoV-2 genome to test the positivity of a sample independently of the variant strain (what we have tried to achieve with our ADESSO; see new Fig. S5e upper section). This test could be then used in combination with an ADESSO-strainX to screen for the presence of a specific variant. This is exactly what we have done now for the detection of Alpha, Beta, Delta and Omicron (see new Figure 3 and related text).

Minor comments

1. What kind of lysis buffer is used for antigen POC test? QuickExtract DNA Extraction solution is not suitable for protein analysis.

Response: We thank the reviewer for pointing this out and we are sorry for the misunderstanding. The antigen test was performed following the manufacturer's instructions as now described in the Methods. QuickExtract was not used in this instance.

2. RT-qPCR can reach 1 copy per μL per CDC EUA, not 2.5

Response: We thank the reviewer for this, we have now changed the text to avoid misunderstanding (lines 102-103).

3. Line 101 "show a lower limit of detection (LoD)" should be "a higher limit of detection (LoD)" or "lower sensitivity".

Response: We have now corrected the statement. We thank the reviewer for noticing this (line 95).

Reviewer #3 (Remarks to the Author):

Casati and colleagues present and validate their method, ADESSO, which is an optimized SHERLOCK-based method for detection of SARS-CoV-2 RNA and specific Alpha and Beta variants. ADESSO can use both extracted or non-extracted nasal and saliva samples as input and has a paper-based lateral flow readout for the ultimate goal of being used in point-of-care (POC) settings. The authors extensively optimize their SARS-CoV-2 assay, which like many SHERLOCK assays uses RT-RPA for amplification and Cas13a for detection, and thoroughly validate their optimized assay on hundreds of patient samples comparing ADESSO's results to RT-qPCR and a rapid antigen test. The authors then demonstrate that this platform can quickly be updated (~1 week) to test for specific variants, a quite valuable feature given the rise of more transmissible variants and/or those that respond differentially to therapies and vaccines.

I first want to applaud the authors for the following strengths of this manuscript:

1. This manuscript is very well written and beautifully organized. The descriptions of the results are clear and easy to follow, overall making this manuscript a joy to read.
2. The inclusion of upstream sample processing methods that can be performed quickly and without extensive hands-on time (i.e. the inclusion of QuickExtract) is an appreciated component of this work particularly for the development of POC tests.
3. The completion of both extensive testing (hundreds of samples) to compare the performance of ADESSO against both RT-qPCR and antigen tests as well as thorough testing of negative samples to set a proper and experimentally validated threshold for calling positive and negative samples, a challenging aspect for visual readouts.

Response: We highly appreciate the positive comments from the reviewer. We have carefully considered all additional comments and suggestions to improve the manuscript.

I have outlined below some major and minor suggestions to improve the manuscript:

Major:

1. I applaud the authors for including references to a majority of the relevant, previously published CRISPR-based SARS-CoV-2 diagnostics in their introduction, but there are a couple concepts that lack sufficient discussion:

a. The authors are correct in stating in Lines 99-101, that many previously developed assays use extracted RNA or have lesser performance on non-extracted samples, but these referenced works use a single reaction set up where amplification and detection are combined in one tube. Although this could be one of the reasons the sensitivity of these assays is reduced, having a single reaction is a huge advantage because (1) the reaction operates at one temperature, (2) less manipulations are required which saves reaction set-up time and assay performance time, and (3) reduces the number of times an amplified sample is manipulated which increases the risk of sample contamination and false positives.

Response: We completely agree with the reviewer and we have now updated the discussion to highlight these aspects (lines 330-333).

b. The authors should discuss their approach in the context of the previously published miSHERLOCK technology (doi: 10.1126/sciadv.abh2944) which applies a one-pot SHERLOCK assay (RPA + Cas12) to detect SARS-CoV-2 variants (both deletions and single-nucleotide changes) using a device that completely automates the assay from sample collection to smartphone-reported results. I think the addition of a Delta

variant assay would better differentiate this manuscript from miSHERLOCK and provide an example of a single-nucleotide variant ADESSO assay.

Response: We thank the reviewer for this suggestion. We now provide data for specific detection of Beta, Delta, and Omicron variant samples based on the detection of single-base mutations (see new Figure 3, S5).

2. The authors should perform some additional technical validation of the saliva-based assay and with the antigen test which could improve or explain some of the differences in sensitivity with this sample type:

a. Figure 1A: Does the QuickExtract method with saliva as input have reduced RNase activity at a similar level as observed with the nasal sample? If not, this could explain the reduced ADESSO LOD for saliva in Figure 1C including some samples with low Ct values and suggest the need for additional units of RNase when testing this sample type.

Response: We have now included the data in the updated Figure 1a. The Reviewer's guess is correct, despite a reduced RNase activity in saliva samples after addition of RNase inhibitor, there is still some RNase activity in comparison to swab samples. We have updated the results accordingly (lines 152).

b. Figure 2C: After quick glance at the product website, it is unclear whether or not the antigen-based test used is validated and compatible with saliva samples. This could explain how only 5 samples were positive by this test and many samples including those with low Ct values failed to test positive. This could potentially be tested by adding SARS-CoV-2 synthetic RNA into a solution with increasing amounts of healthy saliva to see if saliva has an inhibitory effect on the antigen test. I request this instead of retesting patient samples because I know that often patient sample volume is limited.

Response: We thank the reviewer for this comment. The manual of the RIDA®QUICK SARS-CoV-2 Antigen test recommends using nose/throat swabs; they do not mention saliva. Our choice of the antigen test was limited by two main factors, 1) the compatibility with saline solution 0.9% because all our clinical samples were collected in this media (also gargle), and 2) we wanted to use a test that is recognized by the German authority ([https://antigentest.bfarm.de/ords/f?p=1010:100:::~::](https://antigentest.bfarm.de/ords/f?p=1010:100:::)). The RIDA®QUICK SARS-CoV-2 Antigen had both features, therefore we considered it a good representative for the antigen tests commonly used in society.

We would be happy to make the experiment the reviewer proposed but unfortunately, we don't have access to SARS-CoV-2 nucleocapsid protein, which is the element detected during the antigen test. However, we have now updated the results and speculated about the reasons behind such a low sensitivity (lines 211-213).

3. In some instances, there needs to be a bit more clarity regarding ADESSO and direct ADESSO's LOD and sensitivity when compared to other assays:

a. In Figure 1B-D: Clarify whether these assay optimizations and LOD experiments were performed with 2.5 uL of just synthetic RNA or RNA spiked into a mixture of RNase inhibitor, QuickExtract, and a healthy swab sample. It is possible that Quick Extract could impact the performance of both RPA and detection and therefore the sample processing mix should be considered in the context of sample input volume and the LOD of the ADESSO Direct assays.

Response: We thank the reviewer for pointing out this aspect. The experiments shown in Figure 1b-d were performed using synthetic RNA spiked into a mixture of a negative swab sample (collected in saline water), RNase inhibitor, and QuickExtract. We have now updated the text accordingly (lines 154-155).

b. Figure 2B-C: The authors are underselling the performance of their assay compared to RT-qPCR, because their main comparator are the results of the COBAS RT-qPCR which was performed previously and not on the same RNA extract used for ADESSO testing. This could particularly affect samples with high Ct values

which could be more susceptible to freeze-thaw and sample extraction causing variation in the RNA concentration. The side-by-side testing against Tib Molbiol would be a more accurate comparison.

Response: We are grateful to the reviewer for this comment. We now have edited the text in the results to highlight this aspect (lines 201-202).

c. In Figure S5: the authors show that a few of their samples had variation at the RPA binding site in their ADESSO assay. Is it possible that some of this variation could have led to some false negatives within the Figure 2B-C data? The authors should discuss this, include text to describe how their primers and crRNA were designed, and evaluate how much diversity exists within their designed assay amplicon compared to available sequence data.

Response: We are grateful to the reviewer for this insightful comment. This is indeed surely a possibility. We have now included this aspect into results (lines 255-258) and updated Figure 3 and S1b to better explain the crRNA design. Furthermore, we also have included a new bioinformatic analysis in Figure S5e, showing the accumulation of any mutations within the S gene of SARS-CoV-2 from the beginning of the pandemic until November 22nd, 2021. In the same panel we also show a table indicating the percentage of exact match against SARS-CoV2 sequences. This analysis results in a percentage close to 100%, showing that ADESSO is very unlikely to be affected by mutations that have appeared so far in the pandemic. We are confident that this new analysis answers the concerns of the reviewer.

Minor:

1. Including line numbers makes it easier to reference specific lines in the manuscript and adding them in myself could mean some of the line number references are off.

Response: We are sorry for the inconvenience. The new version now has line numbers.

2. Figure 1B: Because neither the RT enzyme amount nor sample input were held constant in the results displayed, it is unclear if additional RT enzyme input with 2.5 uL of sample could have an added benefit on the sensitivity of the assay. It is also worth noting that increased volume of sample inherently should increase assay sensitivity because additional copies are added into the reaction.

Response: The reviewer is right. We have now updated Figure 1b showing that increasing RT enzyme is not beneficial to the sensitivity. We have also added a sentence to note the beneficial effect of increasing the volume (lines 156).

3. Figure 1C: I encourage the authors to take a look at doi:10.1038/s41467-020-19258 where the authors test many RT enzymes +/- RNase H with RT-RPA. The results are quite similar to ones presented here, but a few of the enzymes (e.g. SSIV or Maxima with the inclusion of RNase H) could improve ADESSO's performance.

Response: We thank the reviewer for this comment and we have now included this concept in the discussion (lines 335-336).

4. Figure S3A and C: Missing T and C labels.

Response: We thank the reviewer for noticing this, we have now added the labels.

5. Lines 232-236: Clarify whether there was any confirmatory assay to determine which samples were Alpha versus Beta.

Response: We have now clarified this in the text (lines 244-245). Additionally, we provide the sequencing information for any variant sample in new Supplementary File 1.

6. Line 273: ADESSO -> ADESSO Direct

Response: We thank the reviewer for noticing this, we have now corrected accordingly.

7. Lines 273-274: It might be more compelling to discuss the sensitivity by sample type (NP versus saliva) per my comments above.

Response: We have now completely rewritten the discussion to include all the aspects mentioned by the reviewer.

8. Lines 290-305: I would consider incorporating this into the results section and not the discussion.

Response: As suggested by the reviewer, we have now moved this section in the results (lines 270-284).

REVIEWERS' COMMENTS

Reviewer #1 (Remarks to the Author):

The new Figure 3 is very helpful and clarifies some of the previous confusions. I also applaud the efforts of the authors in constructing methods to detect the variants including the single mutations. I have only three questions for authors to clarify/improve.

- 1) Can authors talk about their strategy and perhaps experimental trials that led to the single nucleotide discrimination? I see that in these cases, authors appear to have several pre-programmed mismatches. What are authors' experiences with the locations of the pre-programmed mismatch? Some are in the seed region but some are not. Authors should specifically state the strategy and discuss the optimization process, if any;
- 2) With regards to samples 6-8, authors identified that they are themselves variants, suggesting that it is important to choose the most conserved region for ADESSO control when detecting S gene variants. Having access to the large collection of sequences, authors should discuss alternative ADESSON sites where the least variation is observed;
- 3) Please add a description in Figure 3 caption what fonts in red and blue represent. It looks like that the red represents deviations from the wild-type strain and blue pre-programmed mismatches.

Reviewer #2 (Remarks to the Author):

The author has carefully responded to my comments with additional data. I'm glad to help improve this manuscript which has the potential for a convenient POC test of a new SARS-CoV-2 variant.

Reviewer #3 (Remarks to the Author):

I thank the authors for addressing my concerns in this revised manuscript. I have no remaining concerns with how the work and results are presented. My only reservation for publication is whether or not there is enough novelty here. As I mentioned previously, there is already work showing simplified sample processing and that these assays can be simplified into fewer reactions (either with one-pot or no amplification, both having minimal impairment of sensitivity), and recent work was published demonstrating Cas13-based delta and omicron variant assays, see <https://doi.org/10.1038/s41591-022-01734-1>.

Response to referees' letter

May 2nd, 2022

Dear Reviewers,

Thank you for reviewing our manuscript entitled "**ADESSO: a rapid, adaptable and sensitive Cas13-based COVID-19 diagnostic platform**" for publication in *Nature Communications*. We are happy to know that you are satisfied with the last version of the manuscript. Your insightful comments and critiques have been essential to notably improving the manuscript.

We provide below answers to the remaining questions of Reviewer #1.

Best regards,

Dr. Riccardo Pecori,
(For the authors)

Response to Reviewers' comments

Reviewer #1 (Remarks to the Author):

The new Figure 3 is very helpful and clarifies some of the previous confusions. I also applaud the efforts of the authors in constructing methods to detect the variants including the single mutations. I have only three questions for authors to clarify/improve.

1) Can authors talk about their strategy and perhaps experimental trials that led to the single nucleotide discrimination? I see that in these cases, authors appear to have several pre-programmed mismatches. What are authors' experiences with the locations of the pre-programmed mismatch? Some are in the seed region but some are not. Authors should specifically state the strategy and discuss the optimization process, if any;

Response: We thank the reviewer for raising this critical point. Our strategy was based on the original SHERLOCK study (Gootenberg et al. 2017, ref. 25), in which the authors achieved point mutation discrimination between different viral strains by designing a crRNA in which the target mutation is in position 3, one additional synthetic mismatch is placed in position 4 or 5, and, finally, a third mutation is present in position 24 (see figure 3 of ref. 25). Therefore, we adapted the same strategy but added synthetic mismatches in the crRNAs where required. This design worked nicely for the crRNAs in ADESSO-Beta and Delta (Fig. 3). However, ADESSO-Omicron required further optimization to achieve single nucleotide discrimination on Alpha samples. Thus, we positioned the target mutation in position 7 of the crRNA, another critical location for Cas13 activity (see figures S8 and S9 of ref. 25). This addition led to successful single nucleotide discrimination (Fig. 3). We have added a few lines to the main text to clarify this point (lines 239-248).

2) With regards to samples 6-8, authors identified that they are themselves variants, suggesting that it is important to choose the most conserved region for ADESSO control when detecting S gene variants. Having access to the large collection of sequences, authors should discuss alternative ADESSON sites where the least variation is observed;

The reviewer is entirely correct. We have now added one sentence to point out the importance of this aspect (lines 273-274).

3) Please add a description in Figure 3 caption what fonts in red and blue represent. It looks like that the red represents deviations from the wild-type strain and blue pre-programmed mismatches.

We thank the reviewer for noticing this. We have now added the required text (lines 789-791).